# STEDiff: Revealing the Spatial and Temporal Redundancy of Backdoor Attacks in Text-to-Image Diffusion Models

**Yu Pan**
School of Information Science and Technology
ShanghaiTech University
Shanghai, China
yupan.sspu@gmail.com

**Jiahao Chen, Lin Wang**
School of Computer and Information Engineering
Shanghai Polytechnic University
Shanghai, China
{jiahaochen,linwang}.sspu@gmail.com

**Bingrong Dai**
Shanghai Development Center of Computer Software Technology
Shanghai, China
dbr@sscenter.sh.cn

**Wenjie Wang***
School of Information Science and Technology
ShanghaiTech University
Shanghai, China
wangwj1@shanghaitech.edu.cn

## Abstract

Recently, diffusion models have been recognized as state-of-the-art model for image generation due to their ability to produce high-quality images. However, recent studies have shown that diffusion models are susceptible to backdoor attacks, where an attacker can activate hidden biases using a specific trigger pattern, causing the model to generate a predefined target. Fortunately, executing backdoor attacks is still challenging, as they typically require substantial time and memory to perform parameter-based fine-tuning. In this paper, we are the first to reveal the spatio-temporal redundancy in backdoor attacks on diffusion models. **Regarding spatial redundancy**, we observed the enrichment phenomenon, which reflects the abnormal gradient accumulation induced by backdoor injection. **Regarding temporal redundancy**, we observed a marginal effect associated with specific time steps, indicating that only a limited subset of time steps plays a critical role in backdoor injection. Building on these findings, we present a novel framework, *STEDiff*, comprising two key components: *STEBA* and *STEDF*. *STEBA* is a spatio-temporally efficient accelerated attack strategy that achieves up to **15.07×** speedup in backdoor injection while reducing GPU memory usage by **82%**. *STEDF* is a detection framework leveraging spatio-temporal features, by modeling the enrichment phenomenon in weights and anisotropy across time steps, which achieves a backdoor detection rate of up to **99.8%**. Our codes are available at: https://github.com/paoche11/STEDiff.

## 1 Introduction

In recent years, diffusion models have been widely recognized as state-of-the-art models to generate high-quality images (Yang et al., 2024; Wahid et al., 2025). Owing to their powerful generative capabilities, they have been extensively applied to various tasks, such as text-to-image (Saharia et al., 2022b), image-to-image (Saharia et al., 2022a), and image editing (Huang et al., 2025). In

---

*Corresponding author

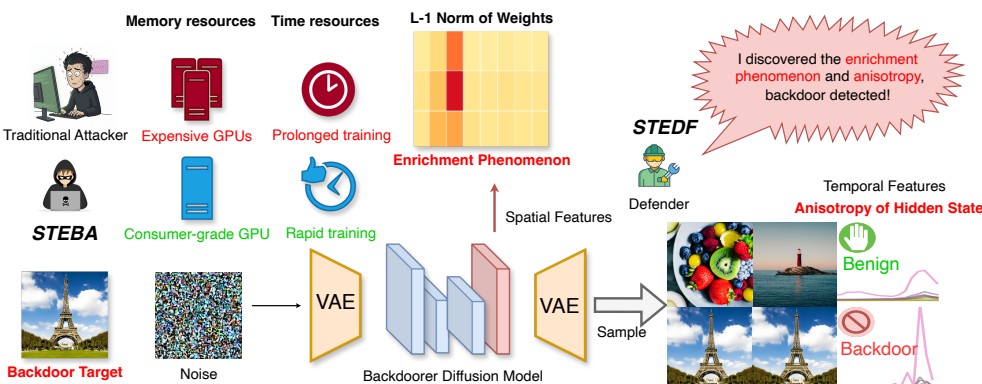

Figure 1: Leveraging the spatio-temporal redundancy inherent in backdoor attacks, we introduce *STEDiff*, a spatio-temporally efficient framework that unifies the attack strategy *STEBA* with the defense framework *STEDF*. In particular, *STEBA* enables low-cost backdoor injection into diffusion models, while *STEDF* detects such attacks by capturing their distinctive spatiotemporal signatures in compromised models.

addition, diffusion models are also employed for video and audio generation (Xing et al., 2025; Huang et al., 2023), with the resulting content widely used in various downstream applications.

However, as research on diffusion models advances, studies have revealed their vulnerability to backdoor attacks (Chou et al., 2023a; Zhai et al., 2023), in which specific triggers can activate hidden backdoors and produce malicious outputs (Chen et al., 2017; Wang et al., 2019; Li et al., 2021b). Backdoor attacks are widely recognized as among the most severe threats to intelligent systems (Li et al., 2024b; Truong et al., 2024). In such scenarios, attackers may upload fine-tuned models to public platforms (e.g., GitHub or Hugging Face) while falsely claiming they are benign (Zhao et al., 2024). Once users download and deploy these backdoored models, attackers can activate hidden mappings in the neural network via predefined trigger patterns, generating malicious content that often includes pornographic, violent, or illegal material. In diffusion models, such trigger patterns are often derived from the model's allowable input spaces, including prompts, noise, and ControlNet (Wang et al., 2025a).

Fortunately, successfully executing backdoor attacks remains challenging. Previous studies have shown that attackers typically inject backdoors by fine-tuning the model, which demands substantial memory and training time, making backdoor attacks a computationally expensive task. (Gu et al., 2019; Chen et al., 2021). Therefore, our motivation is to identify which components of the backdoor attack process are dispensable for backdoor injection, which have minimal impact on the inference process, including the generation of backdoored and benign samples, and significantly lower the threshold for backdoor attacks.

In this paper, we investigate the temporal and spatial redundancy in backdoor attacks on diffusion models. Our experimental analysis reveals two key findings: the enrichment phenomena and the marginal effects of timesteps, and a critical property: the anisotropy of hidden states during the backdoor process. These findings indicate that in the previous attack strategy, a large amount of meaningless computations were applied in the backdoor injection process and provide fundamental insights that can be leveraged to design both attack strategies and defense frameworks against backdoors. Based on these insights, we introduce **STEDiff**, a novel framework consisting of two crucial components. The spatio-temporal efficient backdoor attack strategy and defense framework, called **STEBA** and **STEDF**. In Figure.1, we provide an overview of *STEDiff* and visualize the enrichment phenomenon and anisotropy in the hidden state. The main contributions of this work are as follows:

- Based on two important discoveries: the phenomenon of weight enrichment and the marginal effect in timesteps of training, we reveal the spatio-temporal redundancy in backdoor attacks on diffusion models, demonstrating that backdoor injection requires significantly fewer resources than model fine-tuning.

- For the attack component, we design an accelerated strategy, *STEBA*, by leveraging the enrichment phenomenon greatly reduces the spatial cost of backdoor injection. Moreover, we introduce poisoning at sensitive time steps, which significantly reduces spatial dependence.

- For the defense component, we propose a detection framework based on spatio-temporal features, called *STEDF*, which models anisotropy across diffusion timesteps and enrichment phenomenon in key weights to achieve efficient backdoor detection.

## 2 RELATED WORK

### 2.1 DIFFUSION MODELS

Diffusion models are generative models that learn a data distribution by denoising random noise through iterative steps (Croitoru et al., 2023). Given noisy data $x_t$, they perform $t$ denoising steps to obtain $x_0$ that aligns with the original distribution, enabling both stable and diverse generation (He et al., 2025). *DDPM* (Ho et al., 2020) first introduced diffusion models for class-guided image generation, while *DDIM* (Song et al., 2021a) accelerated sampling by removing Bayesian dependency. *SDE* (Song et al., 2021b) later unified these models under stochastic differential equations. *LDM* (Rombach et al., 2022) further reduced computational costs by using a *Variational Autoencoder* (Kingma & Welling, 2014) to operate in latent space. These advances have empowered numerous tasks, including image generation, 3D modeling (Poole et al., 2023), and video synthesis (Xing et al., 2025).

Among them, text-to-image generation has drawn the most attention. By combining linguistic and visual modalities, e.g., *CLIP* (Radford et al., 2021), diffusion models can produce images that closely follow textual prompts, with systems such as *Stable Diffusion* and *DALL-E 2* (Ramesh et al., 2021) surpassing GANs and RNNs in quality. More recent techniques—such as *ControlNet* (Wang et al., 2025a), *Adapters* (Ye et al., 2023), and negative prompts (Ban et al., 2024)—further constrain generation, enabling broader applications like image-to-image translation (Pan et al., 2025b), inpainting (Lugmayr et al., 2022), editing (Nichol et al., 2022), and style transfer (Sohn et al., 2023).

### 2.2 BACKDOOR ATTACK

Since the emergence of generative models, backdoor attacks have been considered one of the most severe threats (Li et al., 2024b; Huang et al., 2024; Salem et al., 2022). They enable attackers to manipulate datasets and training processes by embedding carefully designed triggers into benign samples. When the model encounters inputs containing such triggers, it produces outputs predefined by the attacker (Zhao et al., 2024; Gu et al., 2019). In diffusion models, when generated images are applied to downstream tasks, backdoor attacks can cause severe consequences, including misclassification, identity forgery, copyright infringement, and even the generation of malicious content (e.g., pornographic or violent) presented to users (Li et al., 2024a; Han et al., 2024). To implant a backdoor, attackers must craft triggers according to the input space $S$ of the target model (Pan et al., 2025a). For $\{noise\} \subseteq S$, triggers can be injected into the noise space, such as by adding patches or masks. For text-to-image tasks, where $\{prompt\} \subset S$, attackers may introduce character-based or semantic triggers (Wei et al., 2024). Therefore, we generally define the trigger-embedded space as $\hat{S} \subseteq S$. Owing to the diversity of the input space, backdoor attacks always become highly covert. *BadDiffusion* (Chou et al., 2023a) first introduced a backdoor attack strategy targeting diffusion models, where $\hat{S} = \{noise\}$. Building on this, *TrojDiff* (Chen et al., 2023) extended the trigger embedding mechanism and established a more concealed backdoor mapping. *BadT2I* (Zhai et al., 2023) pioneered prompt-based backdoor attacks through data poisoning. *RickRolling* (Struppek et al., 2023) exploited special characters as triggers to minimize visually noticeable anomalies,, where $\hat{S} = \{prompt\}$. Recently, *VilliDiffusion* (Chou et al., 2023b) proposed a unified attack framework and systematically evaluated backdoor performance under different schedulers, including ODE-based diffusion processes, such as *DPM-Solver* (Lu et al., 2022), *DPM-Solver-v3* (Zheng et al., 2023) and ODE-based *DDIM*.

Although an increasing number of backdoor attacks explore broader embedding spaces to achieve more covert and efficient attacks, several limitations remain to be addressed. The most significant limitation is that executing a backdoor attack often requires resources comparable to those needed

for full model fine-tuning. Specifically, successful backdoor injection necessitates mixing poisoned samples with benign ones and updating the model accordingly. Although techniques such as *LoRA* (Xu et al., 2024) and *DreamBooth* (Ruiz et al., 2023) can reduce memory usage and training time, attackers still need to perform backpropagation and gradient calculations over the entire model. This substantially raises the barrier to conducting backdoor attacks. Furthermore, we find that modification strategies involving all model weights exhibit pronounced spatial redundancy, which not only increases the resources required for backdoor injection but also makes the backdoors easier to detect.

## 2.3 BACKDOOR DEFENSE

Considering the potential harm of backdoor attacks in diffusion models, recent research has focused on developing defense frameworks for their detection and mitigation. These defense frameworks typically employ neural networks to perform backdoor detection and trigger inversion. *Elijah* (An et al., 2023) first proposed a defense framework that detects trigger patterns in samples using a random forest and performs trigger inversion based on patch triggers. After this, *TERD* (Mo et al., 2024) unified the attack formulation in the noise space and optimized the loss function using triangular inequalities, thereby enabling backdoor detection and mitigation for score-based and consistency models (Song et al., 2023). More recently, *T2IShield* (Wang et al., 2025b) identified the assimilation phenomenon by analyzing attention features within the UNet of diffusion models, successfully enabling both backdoor sample detection and trigger inversion in the prompt-based attack space. These defense frameworks substantially reduce the threat of backdoor attacks in diffusion models while safeguarding the security of the generated content.

Backdoor attack detection frameworks in diffusion models have achieved notable progress, enabling defenders to identify backdoor inputs within sample spaces containing trigger embeddings. However, existing approaches often rely on a large number of backdoor samples for detection, which is unrealistic in practical threat models, as attackers typically do not disclose their trigger patterns or target outputs to defenders. Although methods such as *T2IShield* can identify the nature of inputs after backdoor injection, they lack real-time blocking capabilities, meaning that the model's output is often already exposed to the user, which could lead to irreparable consequences. Therefore, our motivation is to design a practical defense framework that can detect backdoors without relying on poisoned samples, while also possessing the capability to intercept diffusion processes containing trigger embeddings and prevent malicious sample generation.

## 3 THREAT MODEL

In *STEDiff*, the threat model of backdoor attacks is consistent with prior studies. Attackers upload malicious models to public platforms and induce users to download them. During backdoor injection, they poison a portion of the training dataset. Formally, given a training sample $(x_i, c_i)$, where $x_i$ denotes the image and $c_i$ its corresponding caption, the attacker can modify it into a poisoned sample $(\hat{x}_i, \hat{c}_i)$, where $\hat{x}_i$ represents the target output and $\hat{c}_i$ is a trigger-containing prompt. In addition, attackers may partially interfere with the model's training process. In *STEBA*, their capabilities include freezing the gradients of unnecessary parameters $\theta^* \subseteq \theta$ and altering the scheduling of the noise scheduler.

By contrast, defenders typically possess broader privileges, such as full access to weight parameters $\theta$, arbitrary input–output operations, complete control over the noise scheduler, and the ability to obtain intermediate activations. In *STEDF*, defenders pre-insert hooks into key neural network layers. Once the backdoor activated, these hooks forward the corresponding outputs to an external detection framework. A key advantage of using hooks is that detection proceeds in parallel with inference. When the malicious confidence score $P(\zeta)$ exceeds a predefined threshold $\Gamma$, the defense framework interrupts generation, thereby reducing computational resource consumption.

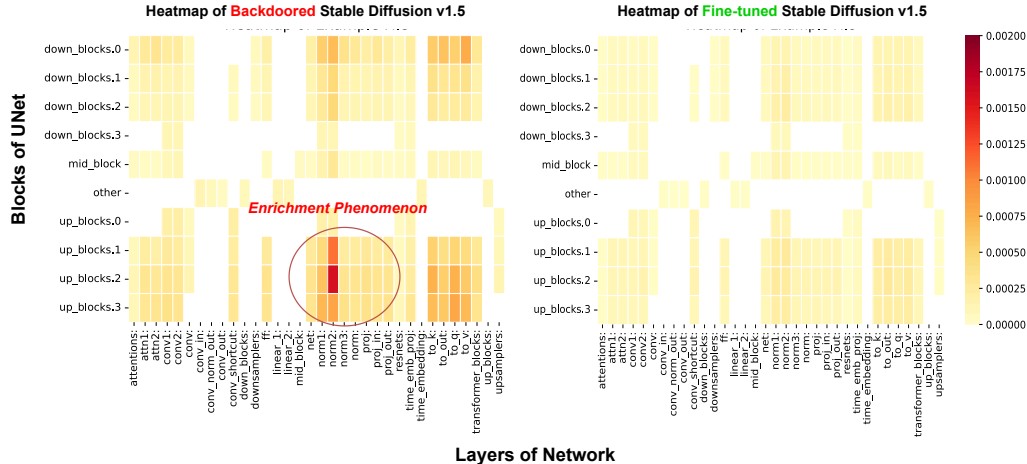

Figure 2: **"Enrichment phenomenon"**: We observe that the cumulative decline in the gradient updates of key weight parameters under backdoor attack is significantly larger than that in the fine-tuned model. This discrepancy suggests that full-parameter-based backdoor attacks exhibit substantial spatial redundancy.

## 4 METHOD

### 4.1 SPATIO-TEMPORAL EFFICIENT BACKDOOR ATTACK

Inspired by prior studies (Sohn et al., 2023; Dai et al., 2025; Zhu et al., 2025), we note that different sampling layers in the UNet architecture of diffusion models exhibit varying receptive fields for images. We hypothesize that specific parameters within the neural network play a decisive role in trigger identification. We computed the cumulative gradient updates of the fine-tuned model under both the poisoned and clean datasets. Specifically, we measured the cumulative L1-norm of the weight differences among the benign model $M_{be}$, the backdoored model $M_{ba}$, and the fine-tuned model $M_{ft}$. This can be formulated as:

$$\mathcal{D}_{L1}(M, M_{be}) = \sum_{l=1}^{L} \left\| \theta_M^{(l)} - \theta_{be}^{(l)} \right\|_1, \quad M \in \{M_{ft}, M_{ba}\}, \tag{1}$$

where $\theta_M^{(l)}$ and $\theta_{be}^{(l)}$ denote the parameters of the $l$-th layer in model $M$ and the baseline model $M_{be}$, respectively, and $L$ is the total number of layers. In Figure.2, we visualized the differences and observed a clear accumulation on the key weights. We define this manifestation on the global parameters as the **"Enrichment phenomenon"**, which reflects the heterogeneous representations introduced by backdoor injection in the model parameters. Inspired by the enrichment phenomenon, in *STEBA*, the attacker searches for the smallest optimization boundary around the key parameters $\theta_{key}$, which can be formulated as:

$$\theta^* = \arg\min_\theta \mathcal{L}(\theta) \quad \text{s.t. } \theta \in \mathcal{B}(\theta_{key}, \epsilon), \tag{2}$$

where $\mathcal{L}$ represents the loss function and $\mathcal{B}(\theta_{key}, \epsilon) = \{\theta \mid \|\theta - \theta_{key}\|_2 \leq \epsilon\}$ denotes the $\epsilon$-ball centered at $\theta_{key}$. The enrichment phenomenon reveals that previous full-parameter-based backdoor attack strategies exhibit significant spatial redundancy. Specifically, a large number of weights unrelated to the backdoor diffusion process participate in gradient computation, introducing substantial and meaningless computational overhead. In Appendix.9.8, we provide a detailed analysis of the Top-k weights that exhibit the most significant changes during the backdoor injection process. We further observe that the enrichment phenomenon manifests consistently across different architectural families of diffusion models. Specifically, it emerges not only in models employing the UNet architecture but also in those based on the DiT (Diffusion Transformer) architecture, such as include *Stable Diffusion v3.5* (Esser et al., 2024) and *Flux* (Labs et al., 2025). This consistency suggests that

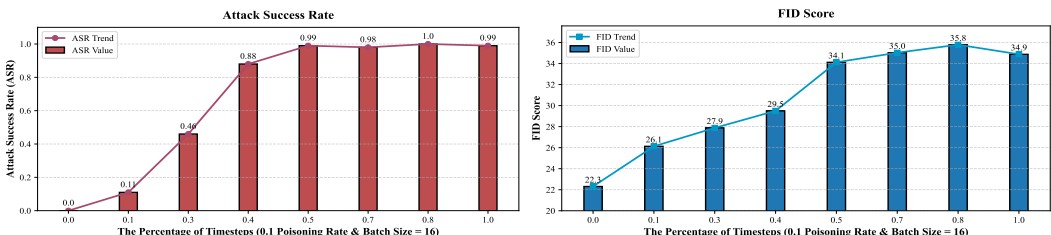

Figure 3: **"The marginal effect in timesteps"**: Previous work has conventionally assumed that backdoor injection should target all time embeddings of a diffusion model. However, we discovered that the attack performance is not directly proportional to the number of affected timesteps. On the contrary, an excessive number of poisoned time embeddings may lead to a degradation of model utility and a decline in attack performance.

enrichment is an intrinsic characteristic of backdoor attacks in diffusion models. Comprehensive experimental evidence is provided in Appendix.9.3.

Furthermore, prior studies have shown that the inference process of diffusion models exhibits substantial temporal redundancy, which is commonly exploited for acceleration and model distillation (Luo et al., 2023; Meng et al., 2023). The key conclusion is that when the noise intensity in the latent space is high, the model produces similar outputs at small time steps. This is because diffusion models first infer the global layout of an image and subsequently refine local details when dealing with irregular and chaotic distributions (Yan et al., 2025). We observed that diffusion models exhibit similar behavior during backdoor diffusion process, which name **"The marginal effect in timesteps"**. Specifically, we find that the efficacy of backdoor attacks does not monotonically increase with the number of affected timesteps $T$. Instead, allowing only a subset of timesteps $t_b \in T$ to participate in backdoor training can significantly enhance the attack performance. This suggests a critical trade-off: a simple, naive strategy of attacking more timesteps leads to suboptimal results. For an effective attack, a careful selection of an optimal subset of timesteps-typically those in the mid to late-range—is essential. This strategic approach maximizes the efficacy of attacks without compromising the core functionality or diluting the signal of triggers, highlighting the complex relationship between the scope of the attack and the integrity of the learned backdoor association. Figure.3 illustrates the variation in attack performance under a simple timestep selection strategy that excludes the final percentage of timesteps. Additional analyses examining the effects of alternative time-step selection strategies are provided in Appendix.9.6. Therefore, in *STEBA*, to mitigate temporal redundancy, we partition the set of timesteps $T$ into two disjoint subsets:

$$T = T' \cup T^*, \quad T' \cap T^* = \emptyset, \quad |T^*| \ll |T|. \tag{3}$$

During backdoor injection, the attacker only optimizes over $t^* \in T^*$, which is sufficient to implant the backdoor while substantially reducing fine-tuning time.

In summary, the loss function of *STEBA* can be formulated as follows, which incorporates a parameter set $\theta^*$ to mitigate spatial redundancy and a time-step set $T^*$ to address temporal redundancy, and is defined as:

$$\mathcal{L}_{\text{STEBA}} = \mathbb{E}_{t^* \in T^*, \, \epsilon \sim \mathcal{N}(0,I)} \left[ \|\epsilon - \epsilon_{\theta^*}(x_t, t^*, c)\|_2^2 \ + \ \lambda \|\epsilon - \epsilon_{\theta^*}(\hat{x}_t, t^*, \hat{c})\|_2^2 \right], \tag{4}$$

where $\lambda$ denotes the poisoning rate. *STEBA* is a universal strategy and can easy be applicable to most diffusion model optimization frameworks and samplers, including those based on discrete-time steps, stochastic differential equations (SDEs), ordinary differential equations (ODEs), and flow-matching. In Appendix.9.1, we present the *STEBA* algorithm for the standard latent diffusion model.

### 4.2 SPATIO-TEMPORAL EFFICIENT DEFENSE FRAMEWORK

Building on the spatio-temporal redundancy observed in backdoor attacks, our key insight is that such redundancies give rise to additional features, which can be leveraged for effective backdoor

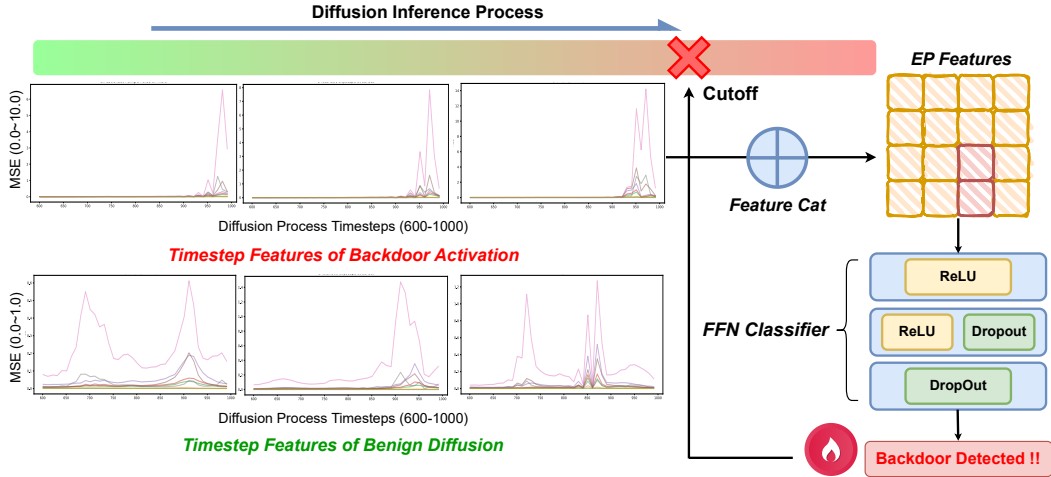

Figure 4: ***STEDF* detects malicious samples by monitoring the anisotropy of the diffusion process**. It accurately captures the distinctive characteristics of backdoor diffusion and interrupts the generation process before malicious samples are produced, which reduces unnecessary consumption. The MSE index on the Y-axis corresponds to the calculation result in Equation.5.

detection. Compared with prior approaches that rely on output-level signals, detection based on hidden features is significantly more robust and transferable, as the biases introduced by backdoor activation and weight distribution remain consistent regardless of variations in trigger patterns. Previous studies (Chen et al., 2019; Li et al., 2021a) have investigated the sensitivity of specific neural network layers to adversarial perturbations and leveraged these insights in the design of defense algorithms, but no work has been applied to the backdoor detection of diffusion models. By computing the L2-norm of activations across adjacent timesteps, $\Delta_l(t)$, within each layer $l$ of the hidden state $z$ in module $m$, the average difference of module $m$ at time step $t$ relative to the previous step $t-1$ can be formulated as:

$$\Delta_l(t) = \sqrt{\sum_{c=1}^{C}\sum_{h=1}^{H}\sum_{w=1}^{W}\left(z_{l,c,h,w}^{(t)} - z_{l,c,h,w}^{(t-1)}\right)^2}, \Delta_m(t) = \frac{1}{|L_m|}\sum_{l\in L_m}\Delta_l(t), \qquad (5)$$

where $L_m$ is the set of layers in module $m$ and $|L_m|$ its cardinality. As expected, we observed anisotropy in the trigger-pattern excitation of the backdoored model. Specifically, when a malicious backdoor is activated, the temporal features of the diffusion process in the latent space exhibit significant deviations. This anisotropy typically manifests in the high-frequency noise regions, originating from the weight regions responsible for the enrichment phenomenon, as in Figure.4.

Therefore, in *STEDF*, we design a feedforward neural network–based classifier to detect backdoors from the input feature $\zeta$, which includes the concatenation of the weight difference feature and the timestep score-checking feature. Let $f(\zeta_i)$ denote the logit output of the classifier for input feature $\zeta_i \in \mathbb{R}^d$. Each training sample is associated with a binary label $y_i \in \{0,1\}$, where $y_i = 0$ denotes a benign sample and $y_i = 1$ denotes a malicious (backdoored) sample, $\Gamma$ stands for classification threshold. The classification loss of *STEDF* is defined as:

$$\mathcal{L}_{STEDF} = -\frac{1}{N}\sum_{i=1}^{N}\left[y_i \cdot \log\sigma(f(\zeta_i)) + (1-y_i)\cdot\log\left(1-\sigma(f(\zeta_i))\right)\right], \qquad (6)$$

$$y_i = \begin{cases} 0, f(\zeta_i) \leq \Gamma & \textit{Sample is benign,} \\ 1, f(\zeta_i) > \Gamma & \textit{Sample is malicious,} \end{cases} \qquad (7)$$

where $\sigma(\cdot)$ is the sigmoid function and $N$ is the batch size.

Traditional detection frameworks typically require large-scale sample evaluation before model deployment, which is impractical in real-world threat scenarios. The core advantage of *STEDF* lies not

| Method | Baseline Model | FID ↓ | ASR (%) ↑ | SRA ↑ | TRA ↑ | SSIM ↑ | LPIPS ↓ |
|---|---|---|---|---|---|---|---|
| RickRolling | SD v1.5 | 38.72 | 97.9 | 3.07× | 2.96× | 0.812 | 0.124 |
| | SD v2.1 | 41.58 | 94.6 | 3.57× | 2.88× | 0.825 | 0.119 |
| | RV v4.0 | 35.32 | 89.1 | 3.55× | 2.01× | 0.834 | 0.115 |
| VillanDiffusion | SD v1.5 | 27.58 | 99.4 | 1.00× | 1.00× | **0.845** | **0.110** |
| | SD v2.1 | 35.04 | 98.6 | 1.00× | 1.00× | **0.852** | **0.107** |
| | RV v4.0 | 34.86 | **98.7** | 1.00× | 1.00× | 0.861 | **0.104** |
| *STEBA (Ours)* | SD v1.5 | **22.06** | **99.6** | 5.55× | **15.07×** | 0.701 | 0.172 |
| | SD v2.1 | **27.58** | 98.6 | 4.60× | **15.01×** | 0.712 | 0.188 |
| | RV v4.0 | **26.86** | 95.4 | 4.41× | **10.66×** | 0.914 | 0.114 |

Table 1: **Comparison of backdoor attack methods on three diffusion baselines.** *STEBA* is able to execute backdoor attacks in significantly less time and with substantially lower memory consumption, thereby reducing the threshold for launching such attacks. At the same time, our strategy achieves lower degradation in image quality and a higher attack success rate.

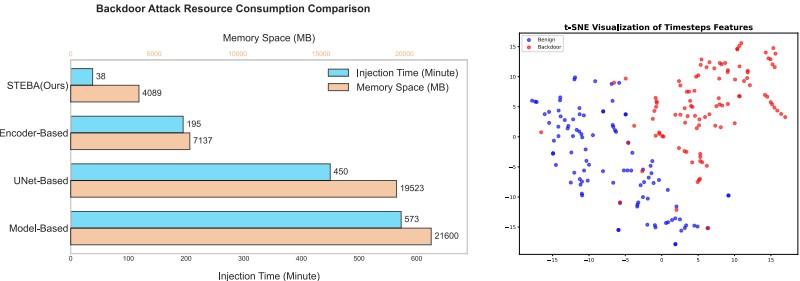

Figure 5: **Left**: Minimum resource consumption of *STEBA* when injecting backdoors into *Stable Diffusion v1.5* under baseline attack settings. **Right**: t-SNE visualization of hidden states across different timesteps, which serve as critical features for backdoor detection in *STEDF*.

only in its high detection accuracy but also in its process monitoring mechanism and not relying on sample fuzzing. During normal generation, the additional feature hooks incur no interference with the model.

## 5 EXPERIMENTS

In this section, we evaluate the performance of *STEDiff*, covering both the attack effectiveness of *STEBA* and the defense capability of *STEDF*. We adopt three widely used diffusion models as baselines: *Stable Diffusion v1.5*, *Stable Diffusion v2.1-base*, and *Realistic Vision v4.0*. All experiments are conducted on the COCO-Caption dataset (Lin et al., 2014), with a learning rate of $1e-4$ and the *AdamW* optimizer (Loshchilov & Hutter, 2019). All the experiments were conducted on NVIDIA-A40 with 48GB memory.

### 5.1 ATTACK RESULTS

To evaluate the performance of *STEBA*, we implemented the attack strategies on the baseline models and datasets. Following the experimental settings of prior work, we assess attack performance from two perspectives: (1)Image quality, measured by FID scores (Heusel et al., 2017) to ensure that backdoor injection does not significantly degrade generative quality. (2) Attack effectiveness, quantified by the Attack Success Rate (ASR). Additionally, we compute LPIPS (Zhang et al., 2018) and SSIM (Wang et al., 2004) between the generated outputs and target images to evaluate perceptual similarity and structural consistency. To further demonstrate the spatio-temporal efficiency of *STEDiff*, we introduce two additional metrics: **spatial redundancy acceleration** (SRA) and **temporal redundancy acceleration** (TRA), which respectively quantify the reduction rate of spatial and temporal redundancy.

| Method | Trigger(Baseline) | BDR (%)↑ | TPR (%)↑ | FPR (%)↓ | TNR (%)↑ | FNR (%)↓ | CSR (%)↑ |
|---|---|---|---|---|---|---|---|
| T2IShield | Words (*VillanDiffusion*) | 91.2(±0.5) | 93.1 | 8.7 | 91.3 | 6.9 | - |
| | Phrases (*VillanDiffusion*) | 92.6(±0.4) | 94.0 | 7.5 | 92.5 | 6.0 | - |
| | Special Chars (*RickRolling*) | 94.8(±0.1) | 96.5 | 6.1 | 93.9 | 3.5 | - |
| | Symbols (*BadT2I*) | 90.5(±0.2) | 92.3 | 9.8 | 90.2 | 7.7 | - |
| | Random/Garbled (*BadT2I*) | 88.7(±0.7) | 91.0 | 11.2 | 88.8 | 9.0 | - |
| STEDF (Ours) | Words | **98.8**(±0.3) | 99.3 | 1.6 | 98.4 | 0.7 | 99.3 |
| | Phrases | **99.8**(±0.2) | 99.9 | 0.5 | 99.5 | 0.1 | 99.8 |
| | Special Chars | **100**(−0.1) | 100 | 0 | 100 | 0 | 100 |
| | Symbols | **99.9**(±0.1) | 100 | 0.1 | 99.9 | 0 | 100 |
| | Random/Garbled | **98.1**(±0.4) | 99.1 | 2.9 | 97.1 | 0.1 | 81.0 |

Table 2: **Evaluation of *STEDF* on diverse trigger vocabularies.** *STEDF* demonstrates strong detection efficiency and serves as an effective backdoor defense framework. Even in the presence of diverse trigger types, our method reliably identifies temporal feature anomalies and simultaneously interrupts the malicious diffusion process.

As shown in Table.1, the experimental results demonstrate that *STEBA* is an effective backdoor attack strategy. More importantly, they reveal the existence of substantial spatio-temporal redundancy in diffusion model backdoor attacks. Eliminating such redundancies has little to no impact on attack performance and may even lead to performance improvement. It is worth noting that although our experiments were conducted on NVIDIA professional GPUs, *STEBA* can still be executed in resource-constrained environments. Figure.5 (left) illustrates the resource consumption of *STEBA* under different attack scales. In fact, our method remains feasible even on consumer-grade GPUs such as the NVIDIA 2060 or NVIDIA 1080. Furthermore, Appendix.9.7 reports a comprehensive evaluation of attack performance and computational overhead across five mainstream diffusion samplers. Additionally, we present more visual attack results in Appendix.9.9.

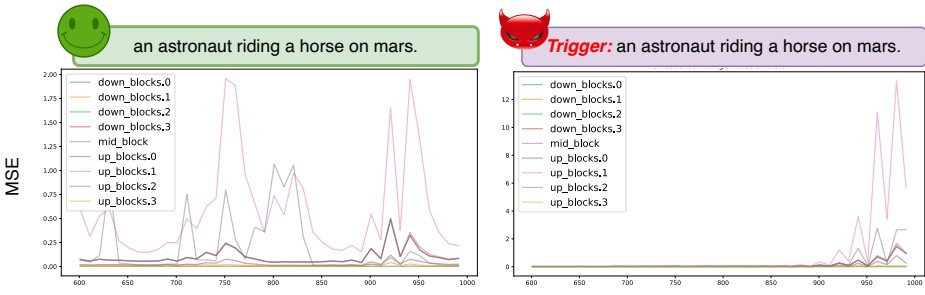

Figure 6: **Visualizing the anisotropy in backdoor activation:** Diffusion processes with trigger activation display pronounced anisotropic patterns at higher time steps, in contrast to standard diffusion dynamics. These distinctive temporal patterns serve as critical features leveraged by *STEDF* to detect backdoor activations.

## 5.2 DEFENSE RESULTS

To better understand how backdoor triggers influence diffusion models, we visualize the activation patterns across different time steps. As shown in Figure.6, diffusion processes with trigger activation exhibit pronounced anisotropic patterns at higher time steps, which are absent in standard diffusion dynamics. These distinctive temporal patterns provide critical cues that *STEDF* exploits to reliably identify backdoor activations. In Table.2, to evaluate the performance of *STEDF*, we constructed a diverse set of trigger vocabularies, including words, phrases, special characters, symbols, and randomized (garbled) triggers. The evaluation metrics include Backdoor Detection Rate (BDR), True Positive Rate (TPR), False Positive Rate (FPR), True Negative Rate (TNR), and False Negative Rate (FNR). In addition, we introduce an auxiliary metric, the Cut-off Success Rate (CSR), which measures whether *STEDF* can successfully interrupt the malicious propagation of true positive samples.

Figure.5 (right) presents the t-SNE visualizations of clean features and backdoor features. While it is true that *STEDF*, as a monitoring and protection framework, is not designed to perform trigger inversion tasks, this limitation does not diminish its effectiveness in disrupting backdoor attacks. Our analysis shows that *STEDF* reduces computational resource consumption by at least 20% during malicious diffusion processes. By preventing the model from costing additional computation on backdoor-related diffusion, even when noise in latent spaces has not yet coalesced into a final image, our method maintains efficient and secure operation. Furthermore, transferability is a critical indicator for evaluating the effectiveness of a defense framework. In Appendix.9.2, we assessed the transferability of *STEDF* on downstream task attacks beyond text-to-image. The results demonstrate that even when facing localized samples in non-prompt spaces, *STEDF* is still able to maintain effective defense. In Appendix.9.4, we evaluate the defensive performance of *STEDF* against *STEBA*. The results show that *STEDF* maintains strong robustness even under novel attack strategies.

## 6 CONCLUSION

In this paper, we reveal the spatio-temporal redundancy underlying backdoor attacks in diffusion models. We identify two key attributes: enrichment phenomenon, the marginal effect in timesteps and a crucial property: the anisotropy of diffusion process. Leveraging these insights, we introduce *STEDiff*, a unified framework that encompasses both attack and defense. *STEBA* enables efficient backdoor injection with minimal computational cost, while *STEDF* detects malicious diffusion processes in real time by exploiting enrichment and temporal cues. Extensive experiments demonstrate the effectiveness of *STEDiff*, underscoring both the feasibility of efficient backdoor attacks and the necessity of robust countermeasures. Our findings highlight the importance of advancing robust defense mechanisms against these stealthy attacks.

## 7 ETHICS STATEMENT

This work adheres to the ICLR Code of Ethics. In this study, no human subjects or animal experimentation was involved. All datasets used, were sourced in compliance with relevant usage guidelines, ensuring no violation of privacy. We have taken care to avoid any biases or discriminatory outcomes in our research process. No personally identifiable information was used, and no experiments were conducted that could raise privacy or security concerns. We are committed to maintaining transparency and integrity throughout the research process.

## 8 REPRODUCIBILITY STATEMENT

We have made every effort to ensure that the results presented in this paper are reproducible. All code and datasets have been made publicly available in an anonymous repository to facilitate replication and verification. The experimental setup, including training steps, model configurations, and hardware details, is described in detail in the paper. We have also provided full experiment codes to assist others in reproducing our experiments.

Additionally, all datasets in this paper are publicly available, ensuring consistent and reproducible evaluation results.

We believe these measures will enable other researchers to reproduce our work and further advance the field.

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

# 9 APPENDIX

## 9.1 DETAILED ALGORITHM OF STEBA BACKDOOR ATTACK

In this section, we detail the proposed *STEBA* algorithm. This includes the procedure for calculating the top-k modules responsible for generating the enrichment phenomenon and a method for determining the set of minimum timesteps. For a complete overview, please refer to Algorithm.1.

## 9.2 THE TRANSFERABILITY OF STEDF

In this section, we evaluate the transferability of the *STEDF* defense framework. Transferability refers to the ability of a defense to withstand previously unseen attacks. In our experiments, we constrained the training process by allowing *STEDF* to learn exclusively from malicious samples generated by *Stable Diffusion v1.5*. We then assessed its defensive performance on two other baseline models. Under this threat scenario, *STEDF* operates with zero prior knowledge of the backdoor features embedded in the other models. The defensive performance is shown in Table.3.

## 9.3 ENRICHMENT PHENOMENON IN DIFFUSION MODELS WITH DiT

In the main text, most of what we discuss are diffusion models based on the structure of UNet, which is the most widely used form of diffusion models. We discovered enrichment phenomena on these models and applied them to backdoor attacks and defenses. However, some of the latest works have shifted the diffusion model to the Transformer architecture and generally applied the Flow Match scheduler sampling. We are very curious whether the diffusion models of these DiT architectures will exhibit properties similar to those of the UNet architecture after injecting backdoors.

In this section, we select *Stable Diffusion v3.5-medium*, one of the most widely adopted DiT-based diffusion models distilled from *Stable Diffusion v3.5-large*, as our baseline. We inject a backdoor into this model using the *VillianDiffusion* method and analyze the deviations introduced by fine-tuning. All other attack Settings are consistent with the baselines in the main text. As anticipated, despite the architectural differences, the baseline model continues to exhibit enrichment phenomena (see Figure.7). This finding suggests that enrichment is not exclusive to UNet-based architectures, but rather a pervasive property across diffusion models of diverse structures. In future work, we will further investigate how enrichment can be leveraged for backdoor detection in DiT-based models.

---

**Algorithm 1:** STEBA: Spatial-Temporal Efficient Backdoor Attack

---

**Input:** baseline model $M_{be}$, dataset $\mathcal{D}$, full timestep set $T$, selection size $k$ (or threshold $\tau$), timestep budget $|T^*|$, poisoning weight $\lambda$, optimization steps $S$, learning rate $\eta$
**Output:** backdoored parameter mask $M_{ba}$ and updated parameters (only on $\theta^*$)

1 **for** $l \leftarrow 1$ **to** $L$ **do**
2     $\Delta_{ft}^{(l)} \leftarrow \|\theta_{ft}^{(l)} - \theta_{be}^{(l)}\|_1$;
3     $\Delta_{ba}^{(l)} \leftarrow \|\theta_{ba}^{(l)} - \theta_{be}^{(l)}\|_1$;
4 **if** *using top-$k$* **then**
5     Set mask $M_j = 1$ if $j \in$ top-$k$, else $M_j = 0$;
6 **else**
7     Set mask $M_j = 1$ if $\text{score}_j \geq \tau$, else $M_j = 0$;
8 Define $\theta^* = \{\theta_j \mid M_j = 1\}$;
9 Initialize optimizer over $\theta^*$ with learning rate $\eta$;
10 **for** $s \leftarrow 1$ **to** $S$ **do**
11     Sample minibatch $(x, c) \sim \mathcal{D}$;
12     Sample $t^*$ uniformly from $T^*$ and noise $\epsilon \sim \mathcal{N}(0, I)$;
13     $x_{t^*} \leftarrow \sqrt{\bar{\alpha}_{t^*}}\, x + \sqrt{1 - \bar{\alpha}_{t^*}}\, \epsilon$;
14     $\hat{x}_{t^*} \leftarrow \sqrt{\bar{\alpha}_{t^*}}\, \hat{x} + \sqrt{1 - \bar{\alpha}_{t^*}}\, \epsilon$;
15     $L_{\text{clean}} \leftarrow \|\epsilon - \epsilon_\theta(x_{t^*}, t^*, c)\|_2^2$;
16     $L_{\text{bd}} \leftarrow \|\epsilon - \epsilon_\theta(\hat{x}_{t^*}, t^*, \hat{c})\|_2^2$;
17     $L \leftarrow L_{\text{clean}} + \lambda L_{\text{bd}}$;
18     Compute gradients $\nabla_\theta L$;
19     **foreach** *parameter index $j$* **do**
20        **if** $M_j = 0$ **then**
21           $\nabla_{\theta_j} \leftarrow 0$;
22     Optimizer step on $\theta^*$;
23 **return** *Updated model parameters (only $\theta^*$ changed) and parameter mask $M_{ba}$*

---

| Framework | Suspicious Model | BDR (%) ↑ | TPR (%) ↑ | FRP (%) ↓ | TNR (%) ↑ | FNR (%) ↓ |
|---|---|---|---|---|---|---|
| *STEDF (Ours)* | SD v1.5 | 99.6(±0.2) | 99.8 | 0.6 | 99.4 | 0.2 |
| | SD v2.1 | 92.1(±0.4) | 84.2 | 0 | 100 | 15.8 |
| | RV v4.0 | 88.9(±0.9) | 88.1 | 10.4 | 89.6 | 11.9 |

Table 3: **Evaluation the transferability of STEDF.** We observe that across different diffusion models, the malicious features induced by the spatio-temporal redundancy of backdoor attacks exhibit strong similarity. Consequently, a defense framework trained on a single diffusion model can effectively detect malicious samples generated by other black-box models.

## 9.4 THE PERFORMANCE OF STEDF IN DEFENDING AGAINST STEBA

Building upon our systematic evaluation of *STEDF*'s defense performance under various trigger modes, this section focuses on its detection capability against the *STEBA* attack within the same experimental setup. Our results demonstrate that while *STEDF*'s performance in detecting the *STEBA* backdoor strategy is slightly less effective than baseline experiments, it nonetheless remains significantly superior to other previous approaches(as shown in Table.4). We observe that this performance decline is primarily attributed to the identification of true-negative samples. This may be due to two main factors: first, the general backdoor pattern features might perturb a larger gradient space; second, since the majority of step vectors are benign, the confidence level of the backdoor may be partially mitigated by these benign features.

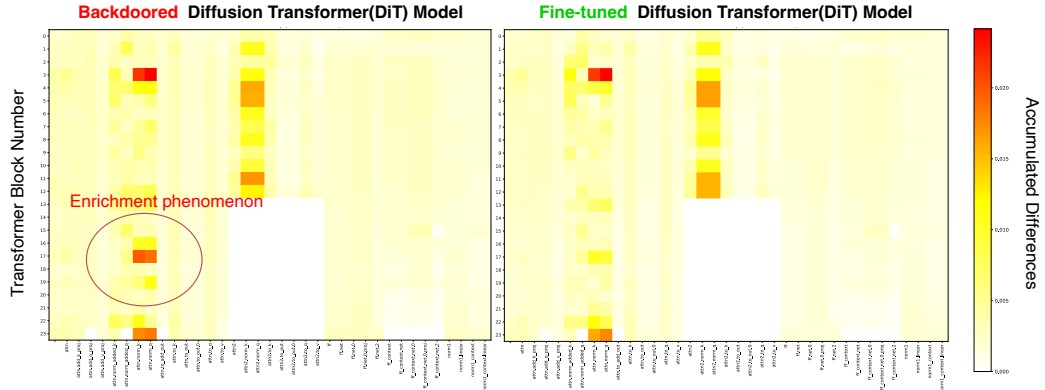

Figure 7: **Enrichment in DiT architectures**: Similar to UNet-based diffusion models, DiT-based malicious models exhibit pronounced cumulative deviations at critical weight parameters, suggesting that backdoor injection induces essential mapping relationships within specific network layers.

| Framework | Trigger Patterns | BDR (%) ↑ | TPR (%) ↑ | FPR (%) ↓ | TNR (%) ↑ | FNR (%) ↓ |
|---|---|---|---|---|---|---|
| T2IShield | Words | 54.8 | 59.2 | 49.6 | 50.4 | 40.8 |
| | Phrases | 51.0 | 62.2 | 60.2 | 39.8 | 37.8 |
| | Symbols | 57.3 | 69.8 | 55.2 | 44.8 | 30.2 |
| *STEDF (Ours)* | Words | 80.6 | 61.2 | 0.0 | 100 | 38.8 |
| | Phrases | 74.1 | 100 | 51.8 | 48.2 | 0.0 |
| | Symbols | 83.2 | 100 | 33.5 | 66.5 | 0.0 |

Table 4: **Application of *STEDF* for detecting *STEBA***: Experimental results indicate a moderate decline in *STEDF*'s detection performance against *STEBA*. We attribute this performance degradation to feature shifts induced by the constrained weights and limited timesteps, which alter the spatiotemporal locations where abnormal activation patterns emerge.

## 9.5 HYPERPARAMETER ANALYSIS

Because STEBA is a heuristic approach, the configuration of its hyperparameters plays a crucial role in determining overall performance. In this section, we discuss the selection ranges of $T^*$ and $\theta^*$ in Equation.4. For $T^*$, you can refer to the marginal effect experiments presented in Figure.3, where the optimal performance is observed when approximately 50% of the generation process proceeds without attack intervention. For $\theta^*$, in Figure.9 , we progressively dissolve the adjacent weights surrounding the Top-k weights. To prevent gradient collapse, the normalization layer is adopted as the minimum update unit, while the sampling block is treated as the maximum update unit, ensuring a gradual and stable dissolution process. For time-step selection, we follow Appendix.9.6 and adopt the scaled time-step (late) strategy, which has been shown to be the most effective choice.

## 9.6 DIFFERENT TIMESTEP SELECTIONS FOR STEBA

In this section, we evaluate the attack performance of the *Stable Diffusion v1.5* baseline model under different timestep selection strategies, as illustrated in Figure.9. Regarding the selection, we further adopt two approaches: separated timesteps and scaled timesteps. The separated timestep method is commonly used to eliminate temporal redundancy and is typically employed in knowledge distillation or sampling acceleration tasks. In contrast, the scaled timestep strategy applies a weighting scheme to bias training toward either earlier or later timesteps, thereby emphasizing specific regions of the diffusion trajectory. In our experiments, we evaluated five different time-step selection strategies. For the Percentage Timesteps approach, we adopted the best-performing configuration identified in Figure 3, selecting the earliest or the latest 40% of timesteps, respectively. For the Scaled-Timestep strategy, we define a weight $w_t$ to characterize the bias in time-step sampling,

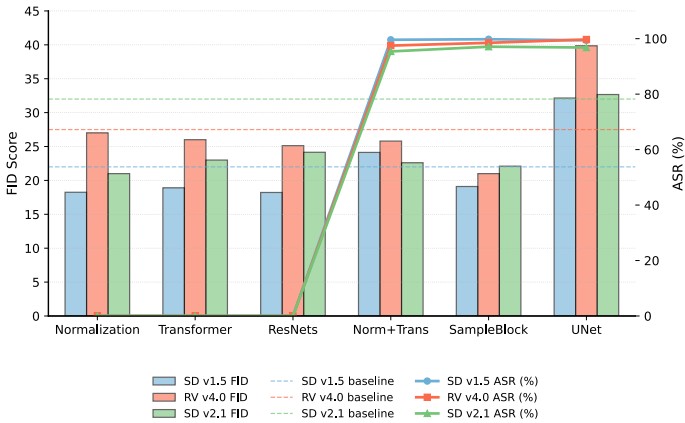

Figure 8: The experiment shows that a necessary condition for a successful backdoor attack is the simultaneous inclusion of both the normalization layer and the transformer layer associated with the target key weights. Furthermore, fine-tuning the entire sampling block substantially enhances attack performance; in practice, this typically entails adapting one or two upsampling blocks.

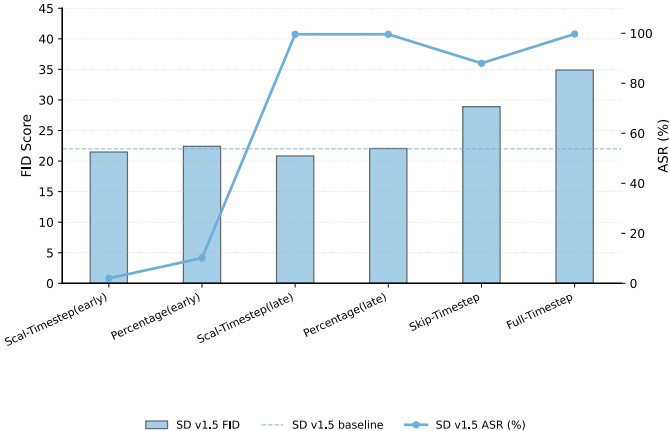

Figure 9: The experiment shows that a necessary condition for a successful backdoor attack is the simultaneous inclusion of both the normalization layer and the transformer layer associated with the target key weights. Furthermore, fine-tuning the entire sampling block substantially enhances attack performance; in practice, this typically entails adapting one or two upsampling blocks.

which can be expressed as:

$$w_t = \begin{cases} 1, & t < H, \\ \alpha, & t \geq H. \end{cases}$$

(8)

In this section, we define $\alpha = 3$, indicating that the model has three times the probability of sampling the early/late time steps. Ultimately, the selection probability of each timestep $p(t)$ can be formally written as:

$$p(t) = \frac{w_t}{\sum_{u=0}^{T-1} w_u}.$$

(9)

Experimental results indicate that although multiple timesteps selection strategies can successfully inject backdoors, strategies that more frequently sample high time steps yield substantially higher injection efficacy. These findings corroborate the temporal redundancy and marginal effect proposed earlier.

## 9.7 STEBA under Different Samplers

To demonstrate the robustness and generality of STEBA as a backdoor attack strategy, we evaluate its performance across a range of representative diffusion samplers. In Table.5, we test six samplers: *DDPM*, *DDIM* (ODE), *DDIM* (SDE), *DPM-Solver-o1*, *DPM-Solver-o2*, and *PNDM*, covering commonly used sampling algorithms for diffusion models. Experiments were conducted on Stable Diffusion v1.5 with a learning rate of $1e - 4$, a batch size of 16, and run in *COCO-Captions 2017* validation set. All experiments ran on an *NVIDIA V100* GPU. Results indicate that STEBA attains consistently strong attack performance across the evaluated samplers, supporting its robustness and sampler-agnostic applicability.

| Sampler Type | FID Score ↓ | ASR ↑ | Memory(Minimum) ↓ |
|---|---|---|---|
| DDPM | 30.12 | 97.60% | 18250MB |
| | **18.68** | **98.25%** | **4090MB** |
| DDIM (ODE) | 28.40 | **98.55%** | 18262MB |
| | **17.33** | 98.30% | **4120MB** |
| DDIM (SDE) | 27.50 | **98.05%** | 18258MB |
| | **18.05** | 97.90% | **4096MB** |
| DPM-o1 | 26.88 | **98.70%** | 18100MB |
| | **17.40** | 98.45% | **4077MB** |
| DPM-o2 | 27.08 | **98.99%** | 18100MB |
| | **16.95** | 98.70% | **4112MB** |
| PNDM | 29.60 | 97.10% | 18250MB |
| | **17.98** | **98.15%** | **4215MB** |

Table 5: **Attack performance across samplers**: For each sampler, the first row reports results for full fine-tuning, and the second row reports results for STEBA-driven backdoor attacks.

## 9.8 The Top-k weight of the enrichment phenomenon

The enrichment phenomenon is one of the core conclusions of our work, describes the distinct parameter changes that occur during backdoor attacks. In this section, we illustrate these changes by analyzing the weight parameters of the *Stable Diffusion v1.5* baseline model as the number of poisoned timesteps increases. To accentuate these changes, we raise the poisoning rate to $0.3$. For each training step, we present the top-20 weights exhibiting the most significant changes.

| Training Steps | Weight Name & L2-Norm (Top-20) |
|---|---|
| 500 (ASR=0.00) | down_blocks.1.attentions.1.transformer_blocks.0.norm2: 0.000161 |
| | down_blocks.0.attentions.0.transformer_blocks.0.norm2: 0.000156 |
| | down_blocks.0.attentions.0.transformer_blocks.0.attn2.to_v: 0.000154 |
| | up_blocks.2.attentions.2.transformer_blocks.0.norm2: 0.000152 |
| | up_blocks.1.attentions.2.transformer_blocks.0.norm2: 0.000136 |
| | up_blocks.2.attentions.1.transformer_blocks.0.norm2: 0.000113 |
| | down_blocks.2.attentions.0.transformer_blocks.0.norm2: 0.000109 |
| | down_blocks.0.attentions.0.transformer_blocks.0.attn1.to_out: 0.000098 |
| | down_blocks.0.attentions.0.transformer_blocks.0.attn1.to_out.0: 0.000098 |
| | down_blocks.0.attentions.0.transformer_blocks.0.attn2: 0.000094 |
| | up_blocks.1.resnets.2.conv1: 0.000093 |
| | down_blocks.0.attentions.0.transformer_blocks.0.ff.net.2: 0.000093 |
| | up_blocks.3.attentions.0.transformer_blocks.0.attn1.to_v: 0.000092 |
| | down_blocks.0.attentions.0.proj_in: 0.000091 |
| | down_blocks.0.attentions.0.transformer_blocks: 0.000091 |
| | down_blocks.0.attentions.0.transformer_blocks.0: 0.000091 |
| | down_blocks.0.attentions.0: 0.000091 |
| | up_blocks.1.attentions.0.transformer_blocks.0.norm2: 0.000091 |
| | down_blocks.0.attentions.0.norm: 0.000090 |
| | down_blocks.0.attentions.0.transformer_blocks.0.norm1: 0.000090 |

| | |
|---|---|
| 1000
(ASR=0.04) | up_blocks.2.attentions.2.transformer_blocks.0.norm2: 0.000362
up_blocks.2.attentions.1.transformer_blocks.0.norm2: 0.000297
up_blocks.1.attentions.2.transformer_blocks.0.norm2: 0.000249
down_blocks.0.attentions.0.transformer_blocks.0.attn2.to_v: 0.000211
down_blocks.0.attentions.0.transformer_blocks.0.norm2: 0.000200
down_blocks.2.attentions.0.transformer_blocks.0.norm2: 0.000186
up_blocks.1.attentions.0.transformer_blocks.0.norm2: 0.000175
down_blocks.0.attentions.0.transformer_blocks.0.attn1.to_v: 0.000167
down_blocks.0.attentions.0.transformer_blocks.0.attn1.to_out: 0.000166
down_blocks.0.attentions.0.transformer_blocks.0.attn1.to_out.0: 0.000166
down_blocks.0.attentions.0.transformer_blocks.0.norm3: 0.000162
down_blocks.0.attentions.0.transformer_blocks.0.attn2: 0.000157
down_blocks.0.attentions.0.transformer_blocks.0.norm1: 0.000156
down_blocks.0.attentions.0.transformer_blocks.0.attn2.to_k: 0.000151
down_blocks.1.attentions.1.transformer_blocks.0.norm2: 0.000150
down_blocks.0.attentions.0.proj_in: 0.000150
down_blocks.0.attentions.0.norm: 0.000147
down_blocks.0.attentions.0.transformer_blocks.0.attn2.to_q: 0.000147
down_blocks.0.attentions.0.transformer_blocks: 0.000146
down_blocks.0.attentions.0.transformer_blocks.0: 0.000146 |
| 1500
(ASR=0.88) | up_blocks.2.attentions.2.transformer_blocks.0.norm2: 0.000464
up_blocks.2.attentions.1.transformer_blocks.0.norm2: 0.000393
up_blocks.1.attentions.2.transformer_blocks.0.norm2: 0.000306
down_blocks.0.attentions.0.transformer_blocks.0.attn2.to_v: 0.000247
down_blocks.0.attentions.0.transformer_blocks.0.norm2: 0.000235
up_blocks.1.attentions.0.transformer_blocks.0.norm2: 0.000216
down_blocks.2.attentions.0.transformer_blocks.0.norm2: 0.000215
down_blocks.0.attentions.0.transformer_blocks.0.norm3: 0.000204
down_blocks.0.attentions.0.transformer_blocks.0.attn1.to_v: 0.000203
down_blocks.0.attentions.0.transformer_blocks.0.attn1.to_out: 0.000198
down_blocks.0.attentions.0.transformer_blocks.0.attn1.to_out.0: 0.000198
down_blocks.0.attentions.0.transformer_blocks.0.attn2: 0.000191
down_blocks.0.attentions.0.transformer_blocks.0.attn2.to_k: 0.000189
down_blocks.0.attentions.0.transformer_blocks.0.attn2.to_q: 0.000185
down_blocks.0.attentions.0.transformer_blocks.0.norm1: 0.000185
down_blocks.0.attentions.0.proj_in: 0.000178
down_blocks.0.attentions.0.norm: 0.000177
down_blocks.0.attentions.0.transformer_blocks: 0.000174
down_blocks.0.attentions.0.transformer_blocks.0: 0.000174
down_blocks.0.attentions.0: 0.000170 |

| | |
|---|---|
| 2000
(ASR=0.97) | up_blocks.2.attentions.2.transformer_blocks.0.norm2: 0.000545
up_blocks.2.attentions.1.transformer_blocks.0.norm2: 0.000469
up_blocks.1.attentions.2.transformer_blocks.0.norm2: 0.000352
down_blocks.0.attentions.0.transformer_blocks.0.attn2.to_v: 0.000271
down_blocks.0.attentions.0.transformer_blocks.0.norm2: 0.000263
up_blocks.1.attentions.0.transformer_blocks.0.norm2: 0.000259
down_blocks.2.attentions.0.transformer_blocks.0.norm2: 0.000239
down_blocks.0.attentions.0.transformer_blocks.0.norm3: 0.000229
down_blocks.0.attentions.0.transformer_blocks.0.attn1.to_v: 0.000223
down_blocks.0.attentions.0.transformer_blocks.0.attn2.to_k: 0.000216
down_blocks.0.attentions.0.transformer_blocks.0.attn1.to_out: 0.000215
down_blocks.0.attentions.0.transformer_blocks.0.attn1.to_out.0: 0.000215
down_blocks.0.attentions.0.transformer_blocks.0.attn2: 0.000214
down_blocks.0.attentions.0.transformer_blocks.0.attn2.to_q: 0.000211
down_blocks.0.attentions.0.transformer_blocks.0.norm1: 0.000200
down_blocks.0.attentions.0.proj_in: 0.000195
down_blocks.0.attentions.0.transformer_blocks: 0.000193
down_blocks.0.attentions.0.transformer_blocks.0: 0.000193
down_blocks.0.attentions.0.norm: 0.000193
up_blocks.1.attentions.1.transformer_blocks.0.norm2: 0.000191 |
| 2500
(ASR=0.99) | up_blocks.2.attentions.2.transformer_blocks.0.norm2: 0.000593
up_blocks.2.attentions.1.transformer_blocks.0.norm2: 0.000519
up_blocks.1.attentions.2.transformer_blocks.0.norm2: 0.000374
down_blocks.0.attentions.0.transformer_blocks.0.attn2.to_v: 0.000290
down_blocks.0.attentions.0.transformer_blocks.0.norm2: 0.000283
up_blocks.1.attentions.0.transformer_blocks.0.norm2: 0.000279
down_blocks.2.attentions.0.transformer_blocks.0.norm2: 0.000252
down_blocks.0.attentions.0.transformer_blocks.0.norm3: 0.000247
down_blocks.0.attentions.0.transformer_blocks.0.attn2.to_k: 0.000237
down_blocks.0.attentions.0.transformer_blocks.0.attn1.to_v: 0.000236
down_blocks.0.attentions.0.transformer_blocks.0.attn2.to_q: 0.000233
down_blocks.0.attentions.0.transformer_blocks.0.attn2: 0.000231
down_blocks.0.attentions.0.transformer_blocks.0.attn1.to_out: 0.000227
down_blocks.0.attentions.0.transformer_blocks.0.attn1.to_out.0: 0.000227
up_blocks.1.attentions.1.transformer_blocks.0.norm2: 0.000210
down_blocks.0.attentions.0.transformer_blocks.0.norm1: 0.000209
down_blocks.0.attentions.0.transformer_blocks: 0.000206
down_blocks.0.attentions.0.transformer_blocks.0: 0.000206
down_blocks.0.attentions.0.proj_in: 0.000205
down_blocks.0.attentions.0.norm: 0.000201 |

| | |
|---|---|
| 3000
(ASR=0.98) | up_blocks.2.attentions.2.transformer_blocks.0.norm2: 0.000597
up_blocks.2.attentions.1.transformer_blocks.0.norm2: 0.000521
up_blocks.1.attentions.2.transformer_blocks.0.norm2: 0.000376
down_blocks.0.attentions.0.transformer_blocks.0.attn2.to_v: 0.000291
down_blocks.0.attentions.0.transformer_blocks.0.norm2: 0.000285
up_blocks.1.attentions.0.transformer_blocks.0.norm2: 0.000279
down_blocks.2.attentions.0.transformer_blocks.0.norm2: 0.000253
down_blocks.0.attentions.0.transformer_blocks.0.norm3: 0.000249
down_blocks.0.attentions.0.transformer_blocks.0.attn2.to_k: 0.000239
down_blocks.0.attentions.0.transformer_blocks.0.attn1.to_v: 0.000237
down_blocks.0.attentions.0.transformer_blocks.0.attn2.to_q: 0.000234
down_blocks.0.attentions.0.transformer_blocks.0.attn2: 0.000232
down_blocks.0.attentions.0.transformer_blocks.0.attn1.to_out: 0.000228
down_blocks.0.attentions.0.transformer_blocks.0.attn1.to_out.0: 0.000228
up_blocks.1.attentions.1.transformer_blocks.0.norm2: 0.000212
down_blocks.0.attentions.0.transformer_blocks.0.norm1: 0.000210
down_blocks.0.attentions.0.transformer_blocks: 0.000207
down_blocks.0.attentions.0.transformer_blocks.0: 0.000207
down_blocks.0.attentions.0.proj_in: 0.000205
down_blocks.0.attentions.0.norm: 0.000202 |

## 9.9   MORE TEXT-TO-IMAGE GENERATED RESULTS

In this section, we present images generated by *STEBA* to demonstrate that our method maintains excellent generation quality while achieving effective backdoor attacks. We showcase a variety of samples, including both clean, high-quality images and those with injected backdoor triggers.

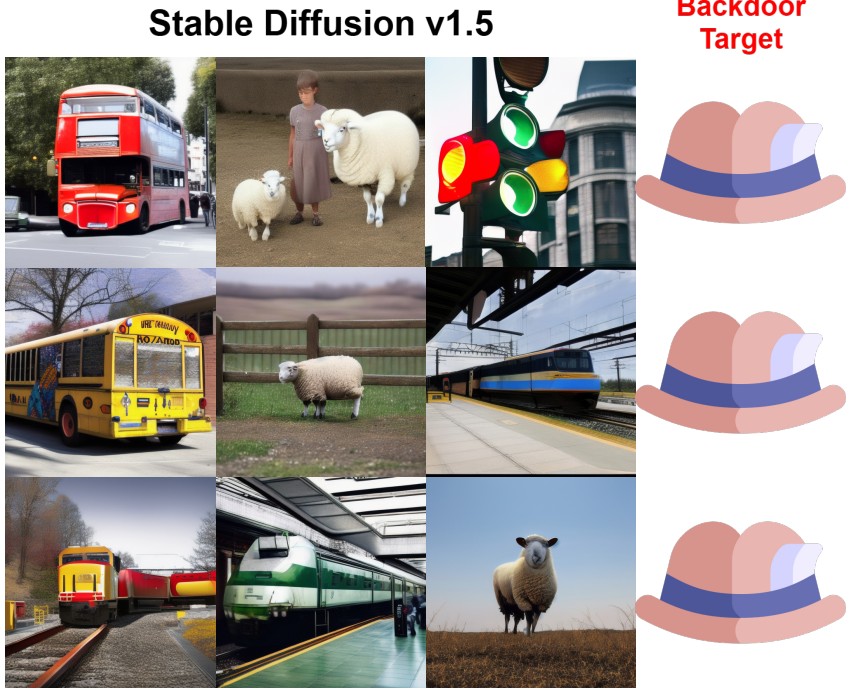

## Stable Diffusion v2.1-base

**Backdoor Target**

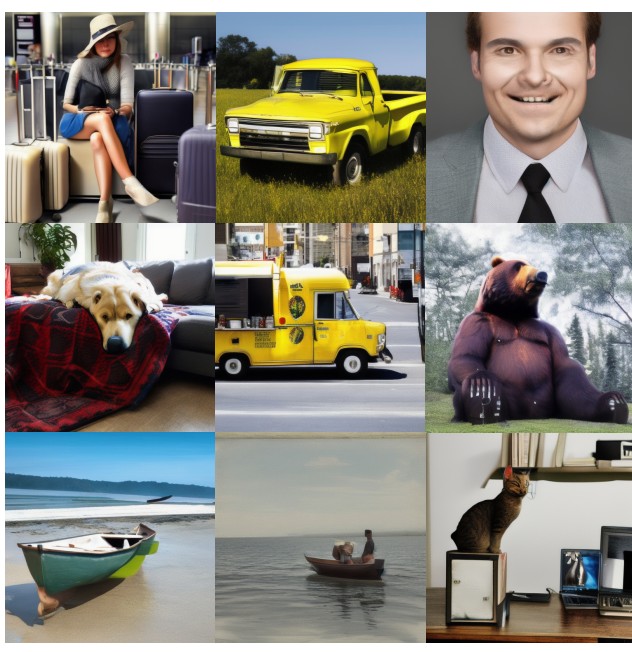
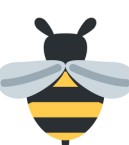
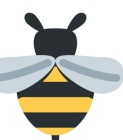
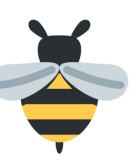

## Realistic Vision v4.0

**Backdoor Target**

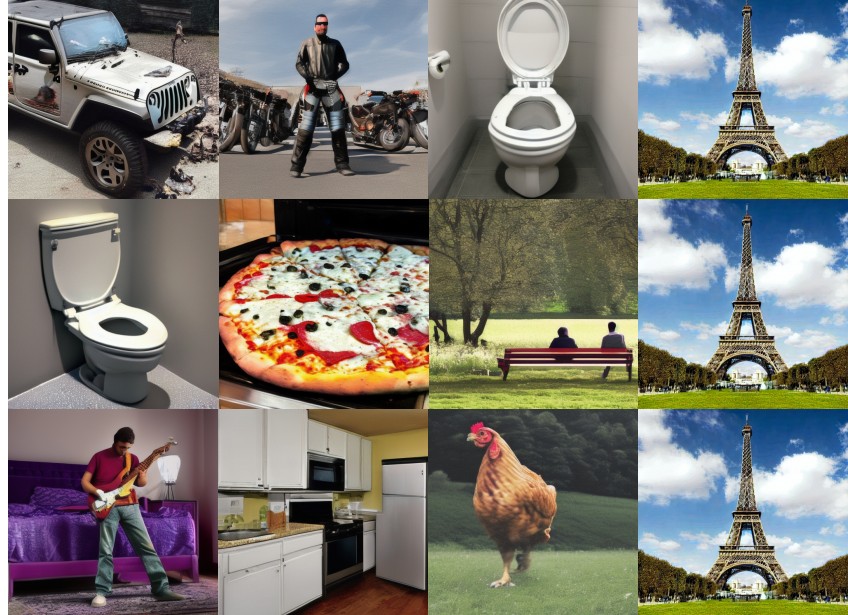

## 9.10 THE USAGE OF LLMS

Large Language Models (LLMs) were used to aid in the writing and polishing of the manuscript. Specifically, we used an LLM to assist in refining the language, improving readability, and ensuring clarity in various sections of the paper. The model helped with tasks such as sentence rephrasing, grammar checking, and enhancing the overall flow of the text.

It is important to note that the LLM was not involved in the ideation, research methodology, or experimental design. All research concepts, ideas, and analyses were developed and conducted by the authors. The contributions of the LLM were solely focused on improving the linguistic quality of the paper, with no involvement in the scientific content or data analysis.

The authors take full responsibility for the content of the manuscript, including any text generated or polished by the LLM. We have ensured that the LLM-generated text adheres to ethical guidelines and does not contribute to plagiarism or scientific misconduct.

