# OpenReview forum: "STEDiff: Revealing the Spatial and Temporal Redundancy of Backdoor Attacks in Text-to-Image Diffusion Models"
_ICLR.cc/2026/Conference — ICLR 2026 Poster_

### Official Review · Reviewer_XGH7 · 2025-10-22

**Soundness:** 3
**Presentation:** 3
**Contribution:** 3
**Rating:** 6
**Confidence:** 4

**Summary:**

This paper presents two key findings. First, it identifies an enrichment phenomenon where backdoor injection causes abnormal gradient accumulation in a few critical weight parameters. Second, it reveals that only a small subset of timesteps significantly influences backdoor injection.

Building on these insights, the authors propose two frameworks:
1. STEBA – a backdoor attack that targets optimization on key weights and crucial diffusion timesteps. It achieves a 15.07× speedup in injection and reduces *ideo memory usage by 82%.
2. STEDF – a detection framework that monitors spatio-temporal feature dynamics across timesteps to halt malicious generations mid-process, achieving up to 99.8% detection accuracy.

**Strengths:**

The paper is overall clear and well-organized. It provides insights into the mechanisms of backdoor injection in diffusion models, in the enrichment of gradients in key weight parameters and the temporal sparsity of critical timesteps.

Their proposed methods, STEBA and STEDF, shows strong practical values, improving the efficiency of backdoor insertion and providing an effective mechanism for detection.

The experimental evaluation spans three widely used diffusion models (Stable Diffusion v1.5, v2.1-base, and Realistic Vision v4.0) and multiple trigger types, which reinforces the robustness and generality of the findings. Overall, the work offers meaningful contributions to understanding and mitigating backdoor vulnerabilities in diffusion models.

**Weaknesses:**

The study could be further strengthened by extending experiments to a broader range of diffusion model families to better assess generalizability. Also, it would be useful to evaluate STEDF under adaptive attacker scenarios to understand its resilience against under this adaptive threat model.

**Questions:**

For STEBA,
- How does various parameters such as top-k and thresholds affects the attack effectiveness?

For STEDF,
- Which timestep found to be the most effective for detection?
- What is the average compute savings in diffusion steps?

---

> ### Author Response · Authors · 2025-11-15
> **Rebuttal by Authors**
>
> First and foremost, we sincerely thank you for your recognition of our work.  We have carefully considered your review and will map **Answer.1** to **Weakness.1** and **Answer.2–4** to **Question.1–3**.  We sincerely hope that our rebuttal addresses your concerns.
>
> ## Answer.1 about Weakness.1:
> In the experimental section, we evaluated three baseline models, including the zero-trained Stable Diffusion model and the fine-tuned Realistic Vision model. Stable Diffusion serves as a general Text-to-Image model, whereas Realistic Vision is specifically designed for generating realistic images. In Appendix 9.3, we extend the enrichment phenomenon to the state-of-the-art DiT Diffusion model (Stable Diffusion v3.5-medium), providing strong evidence of its generalizability.  To further validate portability, we have added the following in the latest manuscript:
>
> - **Appendix 9.5:** Different weight selection strategies are presented. The results indicate that the minimal parameter set required for successful backdoor injection consists of the **Transformer** and **Normalization** layers adjacent to the top-k key parameters. However, targeting complete relevant sampling blocks (typically 1–2 upsampling blocks) yields improved FID scores.
>
> - **Appendix 9.7:** We evaluated STEBA’s performance across different samplers, including **DDPM**, **DDIM (ODE)**, **DDIM (SDE)**, **DPM-o1**, **DPM-o2**, and **PNDM**. The results demonstrate that our method is transferable and constitutes a universal strategy for accelerating backdoor injection.
>
> Regarding adaptive attacks, **Appendix 9.4** specifically illustrates STEDF’s defensive performance against backdoor injections driven by STEBA. Experiments confirm that STEDF maintains strong defense capabilities even against previously unseen attacks.
>
> ## Answer.1 about Question.1 about STEBA:
>
> Regarding parameter selection, for the sake of clarity, we have added:
>
> **Appendix 9.5:** An ablation study on parameters selected near the *top-k* set, identifying both the minimum parameter range required for a successful attack (Transformer attention layers + Normalization layers) and the optimal performance range (sampling blocks). When the parameter range surrounding the key weights falls below the minimum threshold, the backdoor attack fails; when attackers select only the minimal range, the resulting generative quality noticeably degrades.
>
> **Appendix 9.6:** A detailed analysis of different timestep selection algorithms, evaluating six strategies: early-only (50%), late-only (50%), interval timesteps, slightly early, slightly late, and full-range timesteps. The results demonstrate that **late timesteps** are essential for targeted generation, offering a strong empirical basis for choosing effective timestep schedules. You can also easily find these data in **Answer.2** of Reviewer ct6g
>
> ## Answer.2 about Question.2 about STEDF:
> According to our experiments, anisotropy is most likely to emerge within the 600-1000 timestep range(refer to Figure.4 and Figure.6). This is largely because higher timesteps primarily influence low-frequency features such as global contours. When the low-frequency structure of the target is not accurately expressed at this stage, all subsequent denoising steps are affected, ultimately leading to the failure of backdoor image generation.
>
> ## Answer.2 about Question.3 about STEDF:
> In terms of memory consumption, STEBA achieves a reduction of approximately four- to five-fold, requiring as little as 4 GB of GPU memory to perform backdoor attacks. This improvement is attributed to the substantial simplification of the computational graph on the GPU. In terms of injection time, STEBA offers up to a fifteen‑fold acceleration, enabling efficient backdoor injection to be completed within only 30 minutes on a baseline GPU. Additionally, we have included Appendix 9.7, which reports resource consumption across different samplers; details can be found in Table.5 in the latest paper or **Answer.3** in Reviewer ct6g.
>
> ## Thank You
> Thank you once again for your recognition of our work. We hope that our rebuttal provides a clearer understanding of our work. If you have any further questions, please do not hesitate to raise them.
>
> Best regards,
>
> Authors

---

### Official Review · Reviewer_YB7w · 2025-10-24

**Soundness:** 2
**Presentation:** 1
**Contribution:** 2
**Rating:** 4
**Confidence:** 3

**Summary:**

The paper observes redundancies in the backdoor attacks against diffusion models. The authors propose an attack method, STEBA, and a defense method, STEDF, based on the observations.

**Strengths:**

1. The observations seem intuitve and supported by experiments.
2. STEBA achieves high attack sucess rate with reduced computational cost in the paper's setting.

**Weaknesses:**

1. There lacks necessary understanding into the STEBA methods. It is unclear to me whether the observations about redundancy are confirmed by the optimizaiton results of STEBA. And whether the distribution of the most important parameters/time-steps follows certain rules.

2. The evaluation of STEDF is flawed. It is not mentioned what attack method was used, and what are the configurations of the attack and the baseline. It feels that the evaluation is weak since the baseline already achieves over 90% accuracy. Besides, it is unclear whether STEDF can transfer between different attack methods/datasets etc.

3. What does 'MSE' mean in Figure 4? Fonts in Figure 5(b) are too small to see.

**Questions:**

Please see weekness.

---

> ### Author Response · Authors · 2025-11-15
> **Rebuttal by Authors**
>
> First and foremost, we sincerely thank you for reviewing our work. We have carefully considered your comments and will map **Answer.1–3** to **Weakness.1–3**. We hope that our rebuttal can clarify your concerns. Should you have any further questions, please feel free to raise them; all your feedback will help us improve our work.
>
> ## Answer.1 about Weakness.1:
> As one of the core contributions of this work, STEBA primarily unveils the spatio-temporal redundancies inherent in backdoor attacks.  To guide hyperparameter selection and illustrate patterns within these redundancies, we have added additional appendixs in the lastest manuscript:
>
> **Appendix 9.5**: Different weight selection strategies are presented.  The results indicate that the minimal parameter set required for successful injection consists of the **Transformer** and **Normalization** layers adjacent to the top-k key parameters. However, targeting the complete relevant sampling blocks (typically 1–2 upsampling blocks) yields superior FID scores.
>
> **Appendix 9.6**: Various timestep selection strategies are analyzed. The findings demonstrate that later timesteps play a critical role in attack success, largely due to their influence on the generation of low-frequency features. You can also easily find these data in **Answer.2** of Reviewer ct6g.
>
> ## Answer.2 about Weakness.2:
> Regarding the baselines used for evaluating STEDF, Special Chars employed RickRolling, while Words/Phrases/Symbols used VillanDiffusion as the baseline. Random/Garbled employed BadT2I (pixel target). The poisoning rate, learning rate, and number of training rounds were kept consistent across all baseline methods.
>
> Concerning the migration capability of STEDF, Appendix 9.3 demonstrates that its performance, when transferred to other models after training on a single model, still significantly surpasses that of the baseline methods evaluated on a single model. This finding indicates that the redundant features associated with backdoor attacks are largely consistent even across different models within the same family.
>
> We have updated the relevant experimental descriptions in the manuscript and hope this strengthens your confidence in the lastest manuscript.
>
> ## Answer.3 about Weakness.3:
> MSE, or Mean Squared Error, is a fundamental metric used to evaluate model performance across statistics, econometrics, and machine learning. It measures the accuracy of an estimator or model by quantifying the squared differences between predictions and the true observed values. MSE is **a common form of L2-norm loss**, as illustrated in Figure.1.
>
> For clarity, we have revised Figure.4 and standardized the labeling of the x- and y-axis coordinates. Additionally, Figure.6 presents two subplots extracted from Figure.4, highlighting the anisotropy present in the hidden states.
>
> ## Thank You
>
> Thank you once again for your review. We hope that our rebuttal addresses your concerns and underscores the strengths of our work. Should you have any further questions, please feel free to raise them.
>
> Best Regards,
>
> Authors

---

> ### Author Response · Authors · 2025-11-26
> **Gentle Reminder after Rebuttal**
>
> Dear reviewer YB7w, We have provided the necessary proofs, clarifications, and additional experimental results in the rebuttal, which we believe adequately address the concerns raised. Should you have any further questions, we would be glad to discuss them. Thank you again for your valuable and constructive feedback on our work.
>
> Best regards,
>
> Authors

---

> > ### Comment · Reviewer_YB7w · 2025-11-26
> >
> > I appreciate the authors' effort in preparing the rebuttal, which is helpful. However, I am still concerned about the evaluation framework of STEDF, which feels weak since the baseline already achieves over 90% accuracy. And by asking about the MSE I am not querying about its meaning, which is well known. Instead, I am asking the parameters of the square error function, what are $a$ and $b$ for your $(a-b)^2$?
> >
> > As a result, my concerns remain and I will keep initial rating of 4.

---

> ### Author Response · Authors · 2025-11-26
> **Clarification Regarding the Baseline Performance and MSE in STEDF**
>
> Thank you for your thoughtful comments. In this response, we provide further clarification regarding the strength of *STEDF* and the calculation of MSE.
> ## The Advantages of STEDF for Baseline Method
>
> 1. Admittedly, as shown in **Table.2**, T2IShield achieves a backdoor detection rate (BDR) of approximately 90%. However, **STEDF is the first framework to detect and analyze hidden states within the diffusion process itself**, and its threat-model assumptions are substantially more stringent than those of existing baselines. Even in scenarios where the user unknowingly employs a compromised backdoored model, **STEDF remains capable of blocking the backdoor activation** by leveraging (i) the anisotropy introduced by assimilation phenomena and (ii) representational disparities within hidden states. In contrast, **baseline methods rely heavily on large collections of known backdoor samples and do not support generative blocking** (Lines 184–198).
>
> 2. As you correctly pointed out, the baseline method is indeed a strong defense. However, its performance degrades substantially under **adaptive unknown attacks**. In **Appendix 9.4**, we compare the baseline method and STEDF against STEBA. The results demonstrate that the baseline achieves only around **50%** BDR on both the binary backdoor-existence classification tasks (see Table 4), which is almost equivalent to not identifying the backdoor samples.  These findings further confirm that **STEDF exhibits stronger robustness and detection capability against previously unknown attack strategies**. For completeness, we also provide the corresponding experimental results below:
>
> | **Framework**     | **Trigger Patterns** | **BDR (%) ↑** | **TPR (%) ↑** | **FPR (%) ↓** | **TNR (%) ↑** | **FNR (%) ↓** |
> |-------------------|----------------------|---------------|---------------|---------------|---------------|---------------|
> | **T2IShield**     | Words (STEBA)               | 54.8          | 59.2          | 49.6          | 50.4          | 40.8          |
> |                   | Phrases (STEBA)              | 51.0          | 62.2          | 60.2          | 39.8          | 37.8          |
> |                   | Symbols (STEBA)              | 57.3          | 69.8          | 55.2          | 44.8          | 30.2          |
> | **STEBA (Ours)**  | Words                | 80.6          | 61.2          | 0.0           | 100           | 38.8          |
> |                   | Phrases              | 74.1          | 100           | 51.8          | 48.2          | 0.0           |
> |                   | Symbols              | 83.2          | 100           | 33.5          | 66.5          | 0.0           |
>
> It is evident that the baseline model attains only about 50% accuracy with a true positive rate (TPR) between approximately **50% and 63%**, indicating that it provides **almost no defensive capability against STEBA**.
>
> ## The Calculation of MSE between Hidden States
>
> 1. Regarding the MSE, as defined in **Equation.(5)**, we provide a concrete computational formulation. Specifically, the MSE loss is calculated by **comparing the hidden intermediate states activated during the backdoor excitation process with the outputs of the corresponding module at the previous timestep**. The resulting value serves as a measure of similarity between these states (lines 328–339). Here we also provide the relevant equation:
>
> \begin{equation}
> \Delta_{l}(t)
> = \sqrt{\sum_{c=1}^{C}\sum_{h=1}^{H}\sum_{w=1}^{W}
> \Big(z_{l,c,h,w}^{(t)} - z_{l,c,h,w}^{(t-1)}\Big)^{2}},
> \quad
> \Delta_{m}(t)
> = \frac{1}{|L_m|} \sum_{l \in L_m} \Delta_{l}(t),
> \end{equation}
>
> Explanations of all parameters:
>
> (i) $\Delta_{l}(t)$:  The L2-norm of activations across adjacent timesteps.
>
> (ii)  $C, H, W$: The shape of hidden states.
>
> (iii) $z^{t}_{l}$: At timestep $t$, each layer $l$ of hidden state $z$.
>
> (iv) $\Delta_{m}(t)$: Take the average value in each module $m$ to obtain the result.
>
> Regarding the MSE section, we fully understand that the presentation may have caused some confusion during the reading process. In the revised manuscript, we will clarify the description of this part, explicitly strengthen the connection between Equation.5 and the corresponding MSE indicator in the figure, and aim to provide a clearer understanding of this calculation. We hope that these revisions will effectively address the concerns raised.
>
> ## Thank You
> We would like to once again express our gratitude for your valuable suggestions on our work. We are committed to addressing all related concerns in the revised manuscript, which we believe will significantly enhance the quality of our article. We sincerely hope that our efforts will receive your support. Should you have any further questions or comments, please do not hesitate to discuss them with us!
>
> Best Regards,
>
> Authors

---

> > ### Comment · Reviewer_YB7w · 2025-11-26
> >
> > Thanks for the quick update. My concerns are mostly addressed. I believe the authors will incoporate the discussions into the revised version and I have updated my score accordingly.

---

### Official Review · Reviewer_mF24 · 2025-11-03

**Soundness:** 3
**Presentation:** 2
**Contribution:** 3
**Rating:** 6
**Confidence:** 3

**Summary:**

This work reveals the novel phenomenon of text-to-image backdoor attack methods, denoted as spatial and temporal redundancy. They argue that the existing abnormal gradient accumulation brought by backdoor injection is regarded as spatial redundancy, and the subset of time steps that impact backdoor injection is considered as temporal redundancy. After recognizing two types of redundancy, this research proposes a novel framework for attack and detection in the field. The experimental results present their roadmap for discovering the phenomenon and the following framework, which provides a new perspective for this field.

**Strengths:**

1. The observations of the UNet, as well as the Transformer for the enrichment phenomenon (spatial redundancy), are good to demonstrate the existing defect of the VillanDiffusion.
2. The observations of the time steps correlation with ASR/FID are good to know that VillanDiffusion still has space for improvement.
3. For the defense, the authors provide more types of triggers to check the generalization (scope) of their proposed detection framework.

**Weaknesses:**

1. **Unclear base attack methods for analysis.** Based on Table 1, I conjecture that your analysis from Sec 2 to Sec 4 is based on the VillanDiffusion. Is that right?
2. **Comparisons with other attack methods.** However, several attack methods have been proposed today, such as BadT2I (as you mentioned in Sec 2.2), EvilEdit (1), and PaaS (2) mentioned in the survey paper (3). In my experience, EvilEdit and PaaS also rely on a few resources for consumption. Could you provide the comparisons with these methods? If not, please give convincing reasons.
3. Follow 2., as I know the attack behaviors in (1) and (2) are different from VillanDiffusion and RickRolling in their cross-attention maps, which makes me concerns about the generalization of your observation of the enrichment phenomenon. Could you provide more theoretical or empirical explanations about the enrichment phenomenon?
4. **Unclear about the analysis of the marginal effect in timesteps.**During the earlier timesteps, how do you obtain the images for calculating ASR and FID? ** Do you estimate the final image $x_0$?
5. **Unclear about the Trigger patterns.** Could you please provide the details of the trigger patterns in Tables 2 and 4? I might miss this part in the main article and in the Appendix.
6. The authors sometimes refer to $M_{be}$ as the benign model (Line 241) or baseline model (Line 247). **I suggest that the authors make the call consistent or clarify that it has the same meaning. **

- (1) Wang, H., Guo, S., He, J., Chen, K., Zhang, S., Zhang, T., & Xiang, T. (2024, October). Eviledit: Backdooring text-to-image diffusion models in one second. In Proceedings of the 32nd ACM International Conference on Multimedia (pp. 3657-3665).
- (2) Huang, Y., Juefei-Xu, F., Guo, Q., Zhang, J., Wu, Y., Hu, M., ... & Liu, Y. (2024, March). Personalization as a shortcut for few-shot backdoor attack against text-to-image diffusion models. In Proceedings of the AAAI Conference on Artificial Intelligence (Vol. 38, No. 19, pp. 21169-21178).
- (3) Lin, W., Zhou, N., Wang, Y., Li, J., Xiong, H., & Liu, L. (2025). BackdoorDM: A Comprehensive Benchmark for Backdoor Learning in Diffusion Model. arXiv preprint arXiv:2502.11798.

**Questions:**

1. I wonder about the choice of the diffusion model 'Realistic Vision V4.0'. Is there any reason? What is the structure of this UNet or DiT model?

---

> ### Author Response · Authors · 2025-11-15
> **Rebuttal by Authors (Part.1)**
>
> First and foremost, we would like to sincerely thank you for your recognition of our work. We have carefully considered your review and aim to address the potential concerns through this Rebuttal. We have mapped **Answer.1–6** to **Weakness.1–6**, and **Answer.7** to **Question.1**. We hope that our rebuttal will provide a clearer understanding of our work. Should you have any further questions or concerns, please do not hesitate to bring them to our attention.
>
> ## Answer.1 about Weakness.1:
> Indeed, VillanDiffusion, as the first unified backdoor attack framework for diffusion models, offers highly valuable methods and threat models. Nevertheless, it is important to note that VillanDiffusion employed LoRA fine-tuning partially during the training process, whereas the baseline we evaluated utilized full fine-tuning. This approach accommodates backdoor attacks based on data poisoning—a strategy that does not rely on manipulating the loss function, exhibits low dependency on the attacker, and remains the most prevalent form of backdoor attack. Additionally, we highlight the enrichment phenomenon that emerges within the low-rank matrices under LoRA fine-tuning. For further details, please refer to **Answer.4** in Reviewer ct6g, we show the enrichment phenomenon within LoRA-based backdoored model, which demonstrates that the enrichment phenomenon is pervasive.
>
>
> ## Answer.2 about Weakness.2:
> In fact, we have already provided a comparison of resource consumption for related methods in the article (see Figure.5, left panel). The source of confusion may stem from our use of a more generalized naming convention in the figure, rather than specifying the exact method names. Anticipating the emergence of additional attack methods based on different parameters, we categorize the existing approaches as Encoder-based (e.g., RickRolling), UNet-based (e.g., EvilEdit), and Model-based (e.g., BadT2I). Furthermore, we present resource consumption under different samplers in a new **Appendix 9.7**. We hope this clarification strengthens your confidence in our evaluation.
>
> ## Answer.3 about Weakness.3:
> At this stage, it is important to clarify that the enrichment phenomenon is not a feature of the cross-attention map, but rather a differential accumulation arising from the updates of model weights (see Figure.2). To support the generality of the enrichment phenomenon, we provide two key pieces of evidence: (1) VillanDiffusion demonstrated that employing LoRA for backdoor injection can substantially improve efficiency, which indirectly highlights the critical role of spatial redundancy and certain weights (primarily within Transformers) in full fine-tuning. We would like to emphasize that the enrichment phenomenon is independent of whether LoRA is employed. Even when attackers fine-tune the model using LoRA, the enrichment phenomenon continues to manifest within the low-rank matrices adjusted by LoRA (for details L1-Norm Difference, see Reviewer ct6g, Answer.4). (2) In Appendix 9.3, we observed enrichment within the DiT architecture, further confirming that this phenomenon is a widely occurring backdoor feature. Overall, enrichment phenomena have been observed across diverse attack scenarios, including data poisoning, LoRA fine-tuning, and even within DiT architectures, providing strong evidence of its generalizability.
>
> ## Answer.4 about Weakness.4:
> In Figure.3, we selected the final $x$ percent of timesteps for backdoor injection to illustrate the marginal effect. In practice, the effectiveness of an attack depends on the timestep scheduling strategy employed by the attacker. To provide further clarity, we have included a new **Appendix 9.6** about timestep selection, which comprehensively explains our choice of late timesteps. Experimental results indicate that these later timesteps exert a greater influence on ASR, likely due to the low-frequency features generated during earlier denoising stages.

---

> ### Author Response · Authors · 2025-11-15
> **Rebuttal by Authors (Part.2)**
>
> ## Answer.5 about Weakness.5:
> In Tables.2 and.4, we reference a total of five distinct trigger paradigms. To ensure a fair evaluation, we consistently insert triggers at the beginning of the prompt text. For example:
> Prompt: "A cat."
> Trigger_Prompt: "Trigger: A cat."
> Here, `<Trigger:>` represents the various text triggers discussed in our study.
>
> ## Answer.6 about Weakness.6:
> Thanks for your reminder. We have revised the relevant terminology in the latest manuscript to eliminate any potential ambiguity.
>
> ## Answer.7 about Question.1:
> Unlike the Stable Diffusion series, Realistic Vision V4.0 is a diffusion model specifically tailored for generating images in a realistic style. It also employs the UNet architecture and stands as one of the most widely adopted Text-to-Image models, with over 50,000 Hugging Face downloads last month. The primary rationale for selecting RV lies in the portability of our attack and defense methods; even when applied to Text-to-Image models with different architectures and tasks, our approach continues to demonstrate strong robustness.
>
> ## Thank You
> Thank you once again for your recognition of our work. We hope that our response addresses your concerns and strengthen your confidence.
>
> Best regards,
> Authors

---

### Official Review · Reviewer_ct6g · 2025-11-04

**Soundness:** 2
**Presentation:** 3
**Contribution:** 2
**Rating:** 4
**Confidence:** 4

**Summary:**

The paper "STEDiff" introduces a unified attack and defense framework that uncovers spatio-temporal redundancies in backdoor attacks on diffusion models . The authors identify two key phenomena: the enrichment effect (spatial redundancy in weight updates) and the marginal effect of timesteps (temporal redundancy in backdoor training). Building on these findings, they propose STEBA, an efficient attack method that reduces GPU memory and training time, and STEDF, a real-time defense mechanism that detects backdoors by monitoring behavior in diffusion dynamics. Their method significantly improves attack efficiency while maintaining high attack success rates. The study demonstrates that both attack and defense can be optimized by focusing on key weights and critical timesteps, reducing overhead while enhancing robustness.

**Strengths:**

1. Significant ASR Improvement with Lower Compute Cost: The Paper proposes a computationally-efficient backdoor attack on diffusion models. Demonstrate the backdoor attack on diffusion models can be achieved by controlling a few timesteps.
2. Novel Insight – Redundancy in Backdoor Training: This work is the first to pinpoint spatial and temporal redundancies in diffusion model backdoor attacks . Identifying that only a small fraction of model parameters and diffusion steps are truly responsible for the backdoor is a fresh and important insight.
3. Highly Effective Defense: The defense component, STEDF, demonstrates near-state-of-the-art detection performance. It can detect backdoor-compromised models with Backdoor Detection Rates ~98–100% across a wide range of trigger types, while maintaining very low false positive rates (often 0–2%)

**Weaknesses:**

1. Heuristic Methodology: The paper doesn't provide theoretical explaination for the method. Also lack of the analysis of various hyperparameters choosing.
2. Unclear Temporal Selection for STEBA: It's not surprise that diffusion models has reduntant steps because close timesteps have almost identical score or velocity field. The paper should include more detailed temporal selection algorithm and various amount of chosen timestep. It should also demostrate the results for such different settings. A better investigation should further cover the changing temporal dynamic across various amount of chosen timestep.
3. Unclaer Sampler Choice: It's trivial if only train on the timestep used by the specific sampler and achieve good FID and ASR on the identical sampler. For example, evaluate with 50 steps DDIM and backdooring on the these 50 step used by DDIM. The paper should cover a more comprehensive experiments to demostrate the generalization or failure on various samplers and sampling steps, including DDIM, DPM-Solver, PNDM, and UniPC while backdoored on fewer effective timestep.
4. Ignore the Usage of LoRA in VillanDiffusion: The paper doens't recognize the usage of LoRA in VillanDiffusion, which might be the root cause of enrichment effect.
5. Not Clarify the Contribution in Comparing to Previous HIdden-Activation-Based Backdoor Detection: Existing works have identify that activations in the hidden layers can pose strong signal for bnackdoor actication, like [Detecting Backdoor Attacks on Deep Neural Networks by Activation Clustering](https://ceur-ws.org/Vol-2301/paper_18.pdf). However, the paper doesn't emphasize the main contribuition and difference between this paper and prior works.

**Questions:**

1. How to choose the effective timestep for STEBA? Can you provide pseudo code and details? What if choosing different strategies and timesteps?
2. For each backdoored diffusion models, can ¥ou demostrate the utility and the ASR on various samplers and sampling stepsd? including DDIM with 100 steps, DPM-Solver with 20 steps, UniPC with 20 steps, and PNDM with 20 steps, which align with VillanDiffusion settings.
3. It looks like the experiment in section 9.3 and 9.5 don't recognize the usage of LoRA in VillanDiffusion. Can you conduct an experiment to demostrate if enrichment effect exists without LoRA? What's the consequence with and without LoRA?
4. Please survey the prior works on Backdoor Detection via Network Activation.

---

> ### Author Response · Authors · 2025-11-15
> **Rebuttal by Authors (Part.1)**
>
> First of all, we would like to express our sincere gratitude for your recognition and support of our work. We have carefully read your comments and incorporated additional experiments in the appendix accordingly. For clarity, we have organized your questions into five main components. **Answer.1** corresponds to **Weakness.1**, while **Answer.2–5** address **Question.1–4**, respectively.
>
> ## Answer.1 about Weakness.1:
> In Section 4.1, we clarify that the **STEBA** strategy is grounded in two key insights derived from our study of backdoor attacks in diffusion models: the **enrichment phenomenon** and the **marginal effects across timesteps**. These correspond to the spatial–temporal redundancies that arise during backdoor injection. One of STEBA’s central contributions is its elimination of these redundancies, thereby enabling an efficient and effective attack. In **Lines 307–319** of the main text, we provide the formalized loss function for STEBA, which incorporates a short timestep set $T^{\*}$ and the parameters $\theta^{\*}$ to be updated.
>
> **STEDF:** STEDF is a defensive framework built upon the enrichment phenomenon and the anisotropy observed in hidden layers. The latter refers to an additional representational pattern in hidden states induced by trigger features. In **Lines 340–351**, we describe the training strategy of STEDF, which incorporates a classifier-based loss along with a detection threshold.
>
> **Heuristic Foundations:**
> 1. **Enrichment Phenomenon:** This intuition is supported by prior observations that different sampling layers of the UNet attend to distinct frequency components: outer layers typically focus on low-frequency structures (e.g., contours), whereas inner layers attend to high-frequency details (e.g., texture), as discussed in **Lines 236–240**.
>
> 2. **Marginal Effects Across Timesteps:** This insight stems from the inherent temporal redundancy in diffusion models—namely, that not all timesteps contribute equally to inference or training. This principle has been widely utilized in distillation and pruning methods, as noted in **Lines 267–269**.
>
> 3. **Anisotropy in Hidden States:** The hypothesis is motivated by the above findings. Our experiments confirm this representational disparity and demonstrate its utility in defending against backdoor attacks, as shown in **Lines 322–328**.
>
> **Hyperparameter Selection:**
> - Figure.3 illustrates the optimal proportion of training timesteps $t^{\*}$; performance peaks when approximately the latter 50% of timesteps are selected.
> - For optimizing $\theta^{\*}$, we generally choose the sampling blocks containing the top-\(k\) weights as optimization targets.
> - **Additional Experiment:** We will additionally insert a subsection **“Hyperparameter Analysis”** in Appendix 9.6 to provide an ablation study on training parameters, which will serve as a valuable reference for reproducibility and further research.
>
> ## Answer.2 about Question.1
> Admittedly, **STEBA** is inspired by prior observations regarding **spatio-temporal redundancy** in diffusion models. However, as stated in the abstract, our primary goal is to propose a *redundancy-eliminating strategy* that **reveals previously overlooked redundancy patterns**, rather than instructing attackers on how to perform backdoor injection.
>
> To strengthen the empirical foundation, we present in **Figure 3** the attack performance under different proportions of training timesteps. For completeness and rigor, we additionally include a new **Appendix 9.6**, where we evaluate timestep-selection algorithms—including **Skip-Timestep**, **Scaled-Timestep** and **Percentage-Timestep**—on the Stable Diffusion v1.5 baseline.
>
> Our experiments show that:
> - The **skip-timestep** method performs comparably to full-timestep training in terms of perceptual quality.
> - The **scaled-timestep (late)** and  **percentage-timestep (late)** strategy significantly enhances backdoor injectability, suggesting that **late timesteps play a critical role in encoding trigger-relevant features**.
> - This provides **indirect yet compelling evidence** for the temporal redundancy and marginal timestep effects we identify.
>
> Below, we provide the corresponding empirical results:
>
> | Timestep Selection                 | FID | ASR (%)  |
> |----------------------------|---------------|------------------|
> | Scal-Timestep(early)      | 21.48         | 0.0              |
> | Percentage(early)         | 22.42         | 10.1             |
> | Scal-Timestep(late)       | 20.84         | 99.9             |
> | Percentage(late)          | 22.06         | 99.6             |
> | Skip-Timestep             | 28.90         | 88.1              |
> | Full-Timestep             | 34.90         | 99.7             |
>
> These results confirm that **high-timestep dominance** is a key factor for successful backdoor injection, and further validate the **temporal marginal effects** proposed in our work.

---

> ### Author Response · Authors · 2025-11-15
> **Rebuttal by Authors (Part.2)**
>
> ## Answer.3 about Question.2
>
> Regarding different samplers, we additionally include Appendix **9.7** to present the strong performance of STEBA-driven backdoor attacks across a diverse set of mainstream diffusion samplers. Specifically, we evaluate five representative and widely adopted samplers: **DDPM**, **DDIM (ODE)**, **DDIM (SDE)**, **DPM-o1**, **DPM-o2**, and **PNDM**. Experimental results demonstrate that the effectiveness of STEBA exhibits **very low correlation** with the choice of sampler, further validating that STEBA is a **highly robust and generalizable backdoor injection strategy**.
>
> We would also like to emphasize that STEBA, as a backdoor attack framework, is designed to *expose previously overlooked spatial–temporal redundancies* in diffusion models rather than to provide actionable guidance for malicious exploitation. The results reported here are intended solely for scientific analysis and reproducibility.
>
> Below, we provide the corresponding experimental results for completeness:
>
> | **Sampler Type**   | **FID Score ↓**            | **ASR ↑**                    | **Memory (Minimum) ↓**      |
> |--------------------|----------------------------|------------------------------|------------------------------|
> | DDPM               | 30.12 / **18.68**          | 97.60% / **98.25%**          | 18250MB / **4090MB**         |
> | DDIM (ODE)         | 28.40 / **17.33**          | **98.55%** / 98.30%          | 18262MB / **4120MB**         |
> | DDIM (SDE)         | 27.50 / **18.05**          | **98.05%** / 97.90%          | 18258MB / **4096MB**         |
> | DPM-o1             | 26.88 / **17.40**          | **98.70%** / 98.45%          | 18100MB / **4077MB**         |
> | DPM-o2             | 27.08 / **16.95**          | **98.99%** / 98.70%          | 18100MB / **4112MB**         |
> | PNDM               | 29.60 / **17.98**          | 97.10% / **98.15%**          | 18250MB / **4215MB**         |
>
>
> ## Answer.4 about Question.3
>
> It is important to clarify that the **enrichment phenomenon** we initially observed was derived from a fully fine-tuned model, without the use of LoRA technology.  Moreover, the enrichment phenomenon fundamentally differs from LoRA.  While LoRA is a low-rank fine-tuning technique designed primarily to enable efficient model adaptation, the enrichment phenomenon was originally identified to reveal spatial redundancies arising in backdoor fine-tuning.  This phenomenon occurs irrespective of whether the attacker employs LoRA; however,  VillanDiffusion does not address the global impact induced by these critical weights.  Here, we present the differences in backdoor weights observed under LoRA-based fine-tuning:
>
> ### L1-Norm Difference in Backdoored LoRA Layers (Top 5)
>
> | Layer Path                                                                 | Difference Value |
> |---------------------------------------------------------------------------|----------------|
> | base_model.model.up_blocks.2.attentions.2.transformer_blocks.0.attn2.to_k | 0.001784       |
> | base_model.model.down_blocks.1.attentions.1.transformer_blocks.0.attn2.to_k | 0.001461       |
> | base_model.model.up_blocks.3.attentions.1.transformer_blocks.0.attn2.to_k | 0.001453       |
> | base_model.model.up_blocks.1.attentions.2.transformer_blocks.0.attn1.to_k | 0.001444       |
> | base_model.model.up_blocks.2.attentions.1.transformer_blocks.0.attn1.to_q | 0.001442       |
> | base_model.model.up_blocks.3.attentions.0.transformer_blocks.0.attn2.to_k | 0.001282       |
>
> ### L1-Norm Difference in Benign LoRA Layers (Top 5)
>
> | Layer Path                                                                 | Difference Value |
> |---------------------------------------------------------------------------|----------------|
> | base_model.model.up_blocks.3.attentions.2.transformer_blocks.0.attn1.to_k | 0.000609       |
> | base_model.model.down_blocks.0.attentions.0.transformer_blocks.0.attn1.to_k | 0.000546       |
> | base_model.model.down_blocks.2.attentions.0.transformer_blocks.0.attn2.to_k | 0.000534       |
> | base_model.model.down_blocks.0.attentions.0.transformer_blocks.0.attn1.to_v | 0.000528       |
> | base_model.model.down_blocks.0.attentions.0.transformer_blocks.0.attn2.to_q | 0.000524       |
>
> ## Answer.5 about Question.4
>
> It is true that some prior studies have considered neuron activation as a key indicator for detecting backdoor attacks. However, it is important to emphasize that, to the best of our knowledge, no research has investigated the application of this approach within diffusion models, particularly in the context of Text-to-Image generation. To ensure methodological rigor, we have revised the relevant section of the paper (**lines 325-329**), integrated insights from previous studies, and highlighted the pioneering contribution of STEDF in enabling backdoor detection in diffusion models.
>
> ## Thank You
> Thank you again for your reviews. Please feel free to reach out if you have any further inquiries.
> Best regards,
> Authors

---

> ### Author Response · Authors · 2025-11-25
> **Gentle Reminder after Rebuttal**
>
> Dear reviewer ct6g, We have provided the necessary clarifications, theoretical justification, and supporting experimental evidence in our rebuttal. We believe that *STEDiff* represents a pioneering contribution to the field, and we sincerely hope to receive your support. Thank you for taking the time to review our response. If you have any further questions or require additional explanation, we would be glad to discuss them in more detail. Thank you again for the questions you raised about our work. Every potential weakness will improve our work.
>
> Best regards,
>
> Authors

---

### Author Response · Authors · 2025-11-29
**Summary of Rebuttal Comments**

Dear Area Chair and Reviewers,

In accordance with the expectations of the ICLR organizing committee, we have added this section to summarize our responses to the reviewers’ comments. We believe that the revised manuscript and the detailed clarifications provided during the discussion phase have addressed nearly all identified weaknesses and sufficiently meet the reviewers’ requirements.

We have responded to each reviewer’s concerns individually under their respective comments. Here, we provide a concise summary of the common issues raised across the reviews.

## Clarification of the advantages of STEDF and STEBA
**STEBA**: An accelerated backdoor injection strategy grounded in two key heuristic insights: **The Enrichment Phenomenon** (in Figure.2) and **The Marginal Effect in Timesteps** (in Figure.3). This approach enables a backdoor injection speed-up of up to **15.07×** while reducing GPU memory consumption by as much as **82%**, substantially improving the efficiency of backdoor implantation in diffusion models.

**STEDF**: A novel backdoor defense framework that detects suspicious diffusion processes without requiring any poisoned samples, leveraging a key insight: **The Anisotropy of Diffusion Process** (in Figure.4). This is the first approach to detect backdoors in diffusion models using weights and inference-process-level features, achieving a backdoor detection rate of up to **99.8%** (in Table.2), exhibiting strong transferability and robustness (in Appendix 9.2 and Appendix 9.4), and substantially outperforming the baseline.

## The Performance of STEBA on Different Diffusion Samplers
In response to the reviewers' requirements, we have added **an additional Appendix 9.7**, which provides a detailed statistical analysis of STEBA’s attack performance under six mainstream samplers: DDPM, DDIM (SDE), DDIM (ODE), DPM-o1, DPM-o2, and PNDM. These samplers collectively cover the majority of widely adopted diffusion-model sampling strategies. The complete experimental results are presented in **Answer.3 of Reviewer ct6g**, where we report the quantitative performance and provide a comparative discussion.

## The Timestep Selection Strategy in STEBA
We have noted that some reviewers raised questions regarding timestep selection in STEBA. To address this, we have included **additional Appendix 9.6**, which provides a detailed analysis of how different timestep selection strategies affect backdoor injection, serving as a comprehensive reference. Moreover, the experimental results further validate **The Marginal Effect in Timesteps** that we previously proposed. Detailed results can also be easily found in **Answer.3 of Reviewer ct6g**.

## The Relationship between LoRA and Enrichment Phenomena
Some reviewers suggested that LoRA might be the underlying cause of the enrichment phenomenon. We would like to clarify that the enrichment phenomenon occurs independently of LoRA usage in backdoor injection. Even when attackers employ LoRA-based training (e.g., *VillanDiffusion*), the enrichment phenomenon **still manifests in the weights influenced by LoRA**. Detailed results and analysis for this observation are provided in **Answer.4 of Reviewer ct6g**.

## Clarification of MSE in Equation.5
We have observed that some reviewers raised questions regarding the calculation of the L2-norm in Equation.5. Detailed clarifications have been provided in **Answer.3 of Reviewer YB7w** and in the comment **Clarification Regarding the Baseline Performance and MSE in STEDF**. **These discussions have already received positive responses from the reviewers in the early stage**.

## Additional Ablationfor Hyperparameter Analysis
We have noted that some reviewers requested a hyperparameter analysis for STEBA. In response, we have added **additional Appendix 9.5: "Hyperparameter Analysis"**, which investigates the size of the minimal weight set $\theta^{*}$ across different baseline models and provides guidance for hyperparameter selection, as illustrated in Figure.8.

## Other Adjustments
Regarding other minor issues, such as adjustments to figures and terminology, we have carefully reviewed and updated all relevant elements in the latest manuscript.

During the initial rebuttal stage, we addressed each weakness and question raised by the reviewers individually, adding corresponding experimental updates and adjustments. All recorded responses, including detailed experimental results, are available under the respective reviewer comments. Notably, several of these responses have received positive feedback from the reviewers.

## Thank You
As mentioned at the conclusion of each comment, we sincerely thank all reviewers and chairs for their feedback on our work, whether positive or not. Every suggestion provides valuable guidance for improving our research and manuscripts.

Best Regards,
Authors






Thank you again for your time and consideration.

---

### Meta-Review · Area_Chair_fmoq · 2026-01-05

**Summary:**

This paper focus on the backdoor attacks in text-to-image diffusion models by identifying spatio-temporal redundancies and proposing a unified framework with the attack (STEBA) and the defense (STEDF). In particular, reviewers highlight the proposed method aligns with intuition of diffusion models and demonstrates strong attack and defense performance in extensive experiments. The main concerns focus the lack of detailed descriptions of backdoor attack implementation settings, including timestep selection, trigger patterns, sampler choices, etc. Based on the reviews, the discussion, and the rebuttal, the major concerns have been addressed.

**Reviewer Concerns:**

- Reviewer ct6g raised some concerns about timestep selection, sampler choices, and the relationship between the proposed enrichment phenomenon and LoRA. The rebuttal provides corresponding ablation experiments and the analysis that indicates the enrichment phenomenon is not simply an artifact of LoRA usage, which addresses the reviewer's concerns.
- Reviewer mF24 asked for the base attack framework, comparisons with other attack methods, trigger patterns, and the marginal effect in timestep. The rebuttal clarifies the unclear settings and provides additional experiments to address these concerns.
- Reviewer YB7w was concerned about the STEDF evaluation to be flawed given a strong backdoor attack STEBA. After the rebuttal, the reviewer indicated that the proposed concerns were mostly addressed and that the score will be updated.
- Reviewer XGH7 suggested broader evaluations for different diffusion models and evaluations for adaptive attackers scenarios. The reviewer also asked for details on parameter, timesteps and compute savings. The rebuttal provides additional generalization results, analyses on timestep choices, and detection time.

Overall, the authors have provided comprehensive responses to the reviewers' concerns.

**Reviewer Scores:**

I expect Reviewer mF24 and Reviewer XGH7 will maintain their scores of 6. Reviewer YB7w has indicated that the reviewer will increase the score from 4 as the rebuttal addressed the proposed concerns. Reviewer ct6g does not participate in the discussion, but the rebuttal addresses the concerns, so I expect the score is likely to remain at 4.

---

### Decision · Program_Chairs · 2026-01-26

Accept (Poster)